Manuscript prepared for Hydrol. Earth Syst. Sci.
with version 2014/09/16 7.15 Copernicus papers of the LaTeX class copernicus.cls.
Date: 9 November 2016

# Age-ranked hydrological budgets and a travel time description of catchment hydrology

Riccardo Rigon[1], Marialaura Bancheri[1], and Timothy R. Green[2]

[1]Dipartimento di Ingegneria Civile Ambientale e Meccanica, Universitá degli Studi di Trento, Italy
[2]USDA-ARS, Water Management and Systems Research Unit, Fort Collins, Colorado USA

*Correspondence to:* Rigon,R. , Dipartimento di Ingegneria Civile Ambientale e Meccanica,
Universitá degli Studi di Trento, Via Mesiano, 77, 38123, ITALY. (riccardo.rigon@unitn.it)

**Abstract.** The theory of travel time and residence time distributions is reworked from the point of view of the hydrological storages and fluxes involved. The forward and backward travel time distribution functions are defined in terms of conditional probabilities. Previous approaches that used fixed travel time distributions are not consistent with our new derivation. We explain Niemi's formula and show how it can be interpreted as an expression of the Bayes theorem. Some connections between this theory and population theory are identified by introducing an expression which connects life expectancy with travel times. The theory can be applied to conservative solutes, including a method of estimating the storage selection functions. An example, based on the Nash hydrograph, illustrates some key aspects of the theory. Generalisation to an arbitrary number of reservoirs is presented.

## 1 Introduction

Hydrological travel times have been studied extensively for many years. Some researchers (Rodriguez-Iturbe and Valdes, 1979; Rinaldo and Rodriguez-Iturbe, 1996), as reviewed by (Rigon et al., 2016), looked at the construction of the hydrologic response using geographical information. Others (e.g., Uhlenbrook and Leibundgut, 2002; Birkel et al., 2014) used travel times to understand catchment processes in relation to tracer experiments, while new experimental techniques were being developed (e.g., Berman et al., 2009; Birkel et al., 2011). On these premises, (Fenicia et al., 2008; Clark et al., 2011; McMillan et al., 2012; Hrachowitz et al., 2013) aimed to describe both the spatial organization of the catchment and the set of interactions between processes with an assembly of coupled storages (reservoirs) in the number and the organisation necessary to give proper hydrological results without adding unwanted parametric complexity (e.g., Klemeš, 1986; Kirchner, 2006). Despite the simplification efforts, the process of adding physical rigor to their models led to quite complex

systems. Travel time analysis became a tool to disentangle flux complexities (e.g., Tetzlaff et al., 2008), opening the way for explicit unification of geomorphic theories and storage-based modeling (Rigon et al., 2016).

A unique framework for understanding all catchment processes was made possible by the recent work of Rinaldo and others (Rinaldo et al., 2011; Botter et al., 2011). This new branch of research is the focus of the present work. In particular, Botter et al. (2010) and Botter et al. (2011) introduced a newly formulated StorAge Selection function (SAS) related to the probability density function (pdf) of the water age or backward travel-time distribution. With the aid of an apriori assigned
SAS, they were able to write a "master equation" for the travel time probability distribution and solve it, thus systematically connecting the solution of the catchment water budget to travel time aspects of the hydrological flows. Older applications of the travel time theory mostly assumed the simplest case of complete mixing, within the control volume, which is relaxed by using the SAS concept. Subsequently others van der Velde et al. (2012); Benettin et al. (2013, 2015); Harman
(2015b) introduced a new form of the SAS and the age-ranked distribution of water and associated compounds. Firstly, van der Velde et al. (2012) made the SAS a function of the residence time pdfs using actual time, rather than using the "injection time" when water enters the system. Subsequently, Harman (2015b) reformulated the SAS to be a function of the watershed storage and actual time.

      These approaches opened the possibility of exploring the nature of storage-discharge relation-
ships, which are usually parameterised within rainfall-runoff models, and can provide fundamental insight into the catchment functions invoked previously (e.g., Seibert and McDonnell, 2002; Kirchner, 2009). Also the traditional work on groundwater flow and catchment scale transport can be associated with the same ideas, but using time invariant travel time distributions (e.g., Dagan, 1984). Instead, Botter et al. (2011) used an approach that is inherently non-stationary and has immediately
attracted the attention of researchers in that field (e.g., van der Velde et al., 2012; Cvetkovic et al., 2012; Cvetkovic, 2013; Ali et al., 2014). A more detailed history of these concepts can be found in Benettin et al. (2013). Appendix B includes a brief review that is more specifically related to the scope of the present paper. All of these studies provided valuable advances to the theory, but the literature remains obscured by different terminologies and notations, as well as model assumptions
that are not fully explained.

      There remains a need for theoretical developments that are clearly explained and developed using a consistent set of notations. Questions arise, like: does the theory contain hidden parts that are not consistent or explained well? How does it relate to the instantaneous unit hydrograph theory? How can it be used? What generates time varying backward probabilities? Does the theory fully account
for those phenomena which are involved in mobilizing old water (e.g., McDonnell and Beven, 2014; Rinaldo et al., 2015; Kirchner, 2016a)?

      Questions also remain about how to apply the theory of age-ranked distributions in terms of the model form and parameter estimation. Harman (2015a) noted the importance of selecting an appro-

priate SAS, but until very recently (Harman, 2015b), there was no proposed method for selecting the form of an SAS and estimating it from available data. Selection of the SAS for a given watershed remains a topic of importance, since it should not be imposed arbitrarily.

Our work includes a short review of existing concepts that were collected from many (mostly theoretical) papers, which used different conventions and approaches. In the following sections, the theory to date is synthesized into a framework using consistent notation. Besides presenting the concept in a new and organized way, our paper contains some non-trivial answers to the above questions. Clarifications and extensions will be presented and summarized in an integrated manner. These conceptual developments are followed by improved methods for selecting the appropriate form of SAS and estimating its parameters. Guidance for hierarchical approaches to parameter estimation is given, based on available data. Finally, the proposed framework and methods are illustrated using data from an experimental watershed.

## 2 Definitions of age-ranked quantities

Residence time, travel time and life expectancy of water particles and associated constituents flowing through watersheds are three related quantities whose meaning is well defined by the following equation:

$$T = \underbrace{(t - t_{in})}_{T_r} + \underbrace{(t_{ex} - t)}_{L_e} \tag{1}$$

where $T$ [T] ([T] means time units) is the travel time, $t$ [T] is the actual time measured by a clock, $t_{in}$ [T] is the injection time (*i.e.*, the time at which a certain amount of water enters the control volume) and $t_{ex}$ [T] is the exit time (*i.e.*, the time at which a certain amount of water exits the control volume). Based upon these definitions, $T_r := t - t_{in}$ [T] is the so called residence time, or the age of water entered at time $t_{in}$, and $L_e := t_{ex} - t$ [T] is the life expectancy of the same water molecules which are inside of the control volume.

Consider, for example, a control volume as the one shown in figure 1. Its (bulk) water budget is written as:

$$\frac{dS(t)}{dt} = J(t) - Q(t) - AE_T(t) \tag{2}$$

where $S(t)$ [L$^3$] is the time evolution of the water storage, ([L] denotes length units), but instead of volume, we can measure the storage either as mass, or a depth of water [L] (volume per unit area), $J(t)$ [$L^3T^{-1}$] is the precipitation, usually a given (measured) quantity, while the discharge and the actual evapotranspiration, $Q(t)$ [$L^3T^{-1}$] and $AE_T(t)$ [$L^3T^{-1}$], are modeled. Common simple estimates for the two latter quantities are:

$$Q(t) = \frac{1}{\lambda}S^b(t) \tag{3}$$

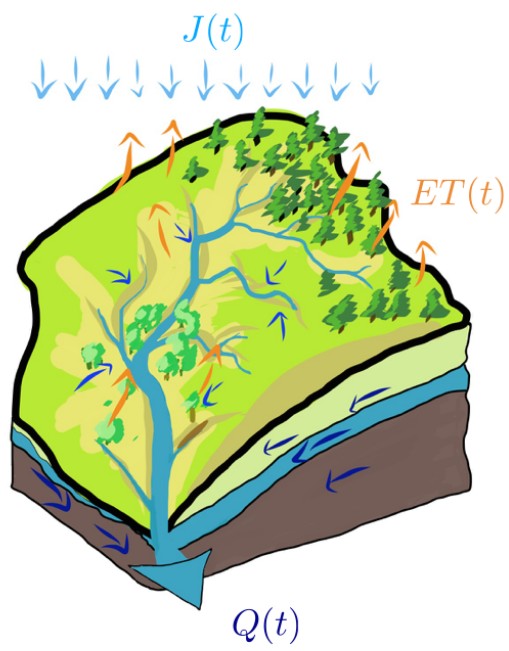

**Figure 1.** A single control volume is considered in which the fluxes are the total precipitation, evapotranspiration and discharge.

and

$$AE_T(t) = \frac{S(t)}{S_{max}}E(t) \tag{4}$$

where $\lambda\,[T]$ and $b$ are the parameters of the non-linear reservoir model, $S_{max}$ is the maximum water storage and $E(t)$ is the potential ET, temporal function of the radiation inputs and atmospheric conditions. Assuming that radiation and various parameters used to model $Q$ and $AE_T$ are given, eq.(2) can be solved and $S(t)$ obtained. If $b = 1$ the budget is a linear ordinary differential equation, and its solution is analytic as in Coddington and Levinson (1955); otherwise, the solution can be obtained through an appropriate numerical solver (e.g., Butcher, 1987). We made the simplification here to use a single storage for illustrative purposes. However, extending the formalism to multiple storages is straightforward, as shown in appendix C.

Being interested in knowing the age of water we need to consider a more general set of equations.

Assume that the water storage $S(t)$ can be decomposed in its sub-volumes $s(t, t_{in})$ [L$^3$ T$^{-1}$] which refer to water injected into the system at time $t_{in} \in [0, t_p]$. Thus:

$$S(t) = \int_0^{\min(t, t_p)} s(t, t_{in})dt_{in} \tag{5}$$

where the initial time $t = 0$ comes before any input into the control volume, and $t_p$ represents the end of the last precipitation considered in the analysis. The variable $t$ represents the actual time at

which the storage is considered. In the following equations, the reference to $t_p$ will be dropped for notational simplicity, and any quantity will consider a limited time interval. The functional form of $s(t, t_{in})$, as well as the functions we define below, can vary with $t$ and $t_{in}$, so they should be labeled appropriately $s_{(t, t_{in})}(t, t_{in})$ but this has been avoided for keeping notations simple.

Analogously, $Q(t)$ [$L^3$ $T^{-1}$] is the discharge out of the control volume, and $q(t, t_{in})$ [$L^3$ $T^{-2}$] is the part of the discharge exiting the control volume at time $t$ composed of water molecules that entered at time $t_{in} \in [0, t_p]$:

$$Q(t) = \int_0^{\min(t, t_p)} q(t, t_{in}) dt_{in} \tag{6}$$

Actual evapotranspiration, $AE_T(t)$ [$L^3$ $T^{-1}$], is the sum of its parts $ae_T(t, t_{in})$ [$L^3$ $T^{-2}$] as:

$$AE_T(t) = \int_0^{\min(t, t_p)} ae_T(t, t_{in}) dt_{in} \tag{7}$$

Finally, let $J(t)$ [$L^3$ $T^{-1}$] denote the input to the control volume. This input can have an "age", and therefore, it can be defined

$$J(t) = \int_0^{\min(t, t_p)} j(t, t_{in}) dt_{in} \tag{8}$$

All these bivariate functions of $t$ and $t_{in}$, $s(t, t_{in})$, $q(t, t_{in})$, and $ae_t(t, t_{in})$ are null for $t < t_{in}$ and can present a derivative discontinuity at the origin ($t = t_{in}$) . Given the above definitions, we can rewrite the water budget as a set of age-ranked budget equations:

$$\frac{ds(t, t_{in})}{dt} = j(t, t_{in}) - q(t, t_{in}) - ae_T(t, t_{in}), \tag{9}$$

These equations were introduced first by van der Velde et al. (2012) and named by Harman (2015a), even though similar ones were present in previous literature, as discussed in Appendix B. In our formulation, however, by using $t$ and $t_{in}$ instead of $t$ and $T_r$ as independent variables, we do not need to transform the original ordinary differential equations (9) into partial differential equations.

## 3 Backward and forward approaches

"Backward" and "forward" are well known concepts in the study of travel time distributions. They were first introduced by Niemi (1977), then by Cornaton and Perrochet (2006), for example, and recently refined by Benettin et al. (2015). Benettin et al. (2015), in particular, related the backward concept to the residence time (or age), while the concept of travel time is both a forward or backward. However, according to us, these previous works didn't fully disclose the inner meaning of the two concepts. In fact, in our theory, the probabilities are defined as backward when they "look" in time to

the history of water molecules and forward when they "look" in time till their exit from the control volume. According to the previous statements, we can define a backward residence time probability, which is conditioned to $t$ and "looks" backward to $t_{in}$, and a forward residence time probability, which is conditioned to $t_{in}$ and "looks" forward to $t$. In the same way, we can define a backward life expectancy probability, which is conditioned to $t_{ex}$ and "looks" backward to $t$, and a forward life expectancy probability, which is conditioned to $t$ and "looks" forward to $t_{ex}$. All these concepts will be clarified further in the following sections.

## 4 Backward Probabilities

Based on the previous definitions, it is easy to define the pdfs of the residence time, travel time and evapotranspiration time. In particular, the pdf of residence time, conditional on the actual time $t$, of water particles in storage, $p_S(T_r|t)$, can be defined as:

$$p_S(T_r|t) \equiv p_S(t - t_{in}|t) := \frac{s(t, t_{in})}{S(t)} \ [\text{T}^{-1}] \tag{10}$$

where "$\equiv$" means equivalence, and ":=" a definition. Benettin et al. (2015) denoted $p_S(T_r|t)$ as $\overleftarrow{p_S}(T_r, t)$ but since this probability density is conditional to the actual time, standard probability notation is clear and unambiguous.

It is evident that this probability is time variant, since the integral and the integrand in equation (5) keep a dependence on the clock time $t$.

The pdf of travel time is $p_Q(t - t_{in}|t)$, where $t_{ex} = t$, since we are considering the water exiting the control volume as discharge. It can be defined as:

$$p_Q(t - t_{in}|t) := \frac{q(t, t_{in})}{Q(t)} \ [\text{T}^{-1}], \tag{11}$$

This definition for the probability is very restrictive, and can imply inconsistencies in those papers which assume a time invariant backward distribution to obtain tracers concentration, as shown in Appendix D. Eventually, the pdf of travel time for water exiting the control volume as water vapor, $p_{E_T}(t - t_{in}|t)$, can be defined as:

$$p_{E_T}(t - t_{in}|t) := \frac{ae_T(t, t_{in})}{AE_T(t)} \ [\text{T}^{-1}], \tag{12}$$

It is also possible to define the mean age of water for any of the two outlets, which is given by:

$$\langle T_r(t) \rangle_i = \int_0^{\min(t, t_p)} (t - t_{in}) p_i(t - t_{in}|t) dt_{in} \tag{13}$$

for $i \in \{Q, E_T\}$, which is a function of the sampling time (and the rainfall input).

After the above definitions, the age-ranked equation (9), can be rewritten as:

$$\frac{d}{dt}[S(t)p_S(T_r|t)] = J(t)\delta(t - t_{in}) - Q(t)p_Q(t - t_{in}|t) - AE_t(t)p_{E_T}(t - t_{in}|t) \tag{14}$$

when a single "new water" injection of mass is considered, and the bulk quantities $S(t)$, $Q(t)$, $AE_T(t)$ are known as soon as the bulk water budget, equation (2), is solved. $\delta(t-t_{in})$ is a Delta-dirac function to account for the water particles in precipitation with age zero. The travel time probabilities on the right side of (14) are not known. Consequently Botter et al. (2011) introduced a storage selection function, $\omega(t, t_{in})$ [-], for each of the outputs, so that:

$$p_Q(t - t_{in}|t) := \omega_Q(t, t_{in})p_S(T_r|t) \qquad (15)$$

and:

$$p_{E_T}(t - t_{in}|t) := \omega_{E_T}(t, t_{in})p_S(T_r|t) \qquad (16)$$

Therefore equation (14), after the proper substitutions, becomes:

$$\frac{d}{dt}[S(t)p_S(T_r|t)] = J(t)\delta(t - t_{in}|t) - Q(t)\omega_Q(t, t_{in})p_S(T_r|t) - AE_t(t)\omega_{E_T}(t, t_{in})p_S(T_r|t) \qquad (17)$$

Once assigned the $\omega(t, t_{in})$ values on the basis of some heuristic, as in Botter et al. (2011), equation (17) represents a linear ordinary differential equation and can be solved exactly as:

$$p_S(T_r|t) = e^{-\int_{t_{in}}^{t} g(x,t_{in})dx}\left[p(0|t) + \int_{t_{in}}^{t} \frac{J(y)\delta(y - t_{in})}{S(y)}e^{\int_{t_{in}}^{t} g(x,t_{in})dx}dy\right] \qquad (18)$$

where :

$$g(x, t_{in}) = \frac{1}{S(x)}\left[\frac{dS(x)}{dt} + Q(x)\omega_Q(x, t_{in}) + AE_t(x)\omega_{E_T}(x, t_{in})\right] \qquad (19)$$

and $p(0|t)$ is the initial condition. This is only valid if equation (17) is linear, i.e. $\omega(t, t_{in})$ is not a function of $p_S(T_r|t)$. Figure 2 shows the variation of the $p_S(T_r|t)$ with the injection time, while the chronological time is kept fixed. The curves were obtained considering three different injections at $t_{in_1}$, $t_{in_2}$ and $t_{in_3}$, and assuming $\omega_Q(t, t_{in}) = \omega_{E_T}(t, t_{in}) = 1$.

The conditional probability $p_S(T_r|t)$ properly integrates to one, as shown in figure 3, when it is
integrated in $t_{in}$. In particular, figure 3 shows that $p_S(T_r|t) = const$, when $J(t) = 0$. In fact, if we consider $\omega_Q(t, t_{in}) = \omega_{E_T}(t, t_{in}) = 1$, equation (17) is simplified in:

$$\frac{d}{dt}[S(t)p_S(T_r|t)] = -Q(t)p_S(T_r|t) - AE_t(t)p_S(T_r|t) \qquad (20)$$

and, therefore,

$$\frac{dp_S(T_r|t)}{dt} = -\frac{p_S(T_r|t)}{S(t)}\left[\frac{dS(t)}{dt} - Q(t) - AE_t(t)\right] = 0 \qquad (21)$$

Figure 4 shows the evolution of $p_S(T_r|t)$ with the actual time $t$ and the injection time kept fixed. The integral of the area under the three curves, obtained for the same three injections, in this case, is not equal to 1, since the functions are not pdfs in $t$.

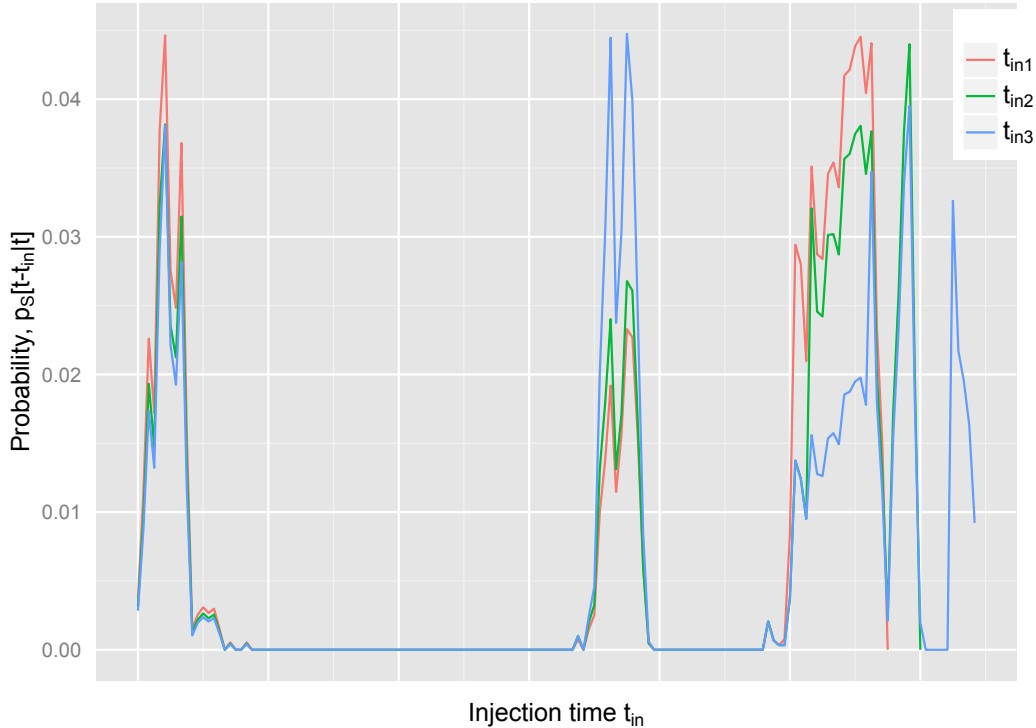

**Figure 2.** Representation of the evolution of the backward pdf for three injection times ($t_{in_i}$, where i = 1,3) as varying with the injection time $t_{in}$. The time shift between the three injections was dropped for a direct comparison of the curves.

## 5  Forward Probabilities

Consider again the age-ranked equation (9). Since we want to track the evolution of a water particle while crossing the catchment, we can write its integral form over dt, as:

$$s(t,t_{in}) = J(t_{in}) - \int_0^t q(t,t_{in})dt - \int_0^t ae_T(t,t_{in})dt \tag{22}$$

It can be rewritten as a probability conditional to $t_{in}$:

$$P_S[t-t_{in}|t_{in}] := 1 - \frac{s(t,t_{in})}{J(t_{in})} = \frac{V_Q(t,t_{in})}{J(t_{in})} + \frac{V_{E_T}(t,t_{in})}{J(t_{in})} \tag{23}$$

having defined:

$$V_Q(t,t_{in}) := \int_0^t q(t,t_{in})dt \tag{24}$$

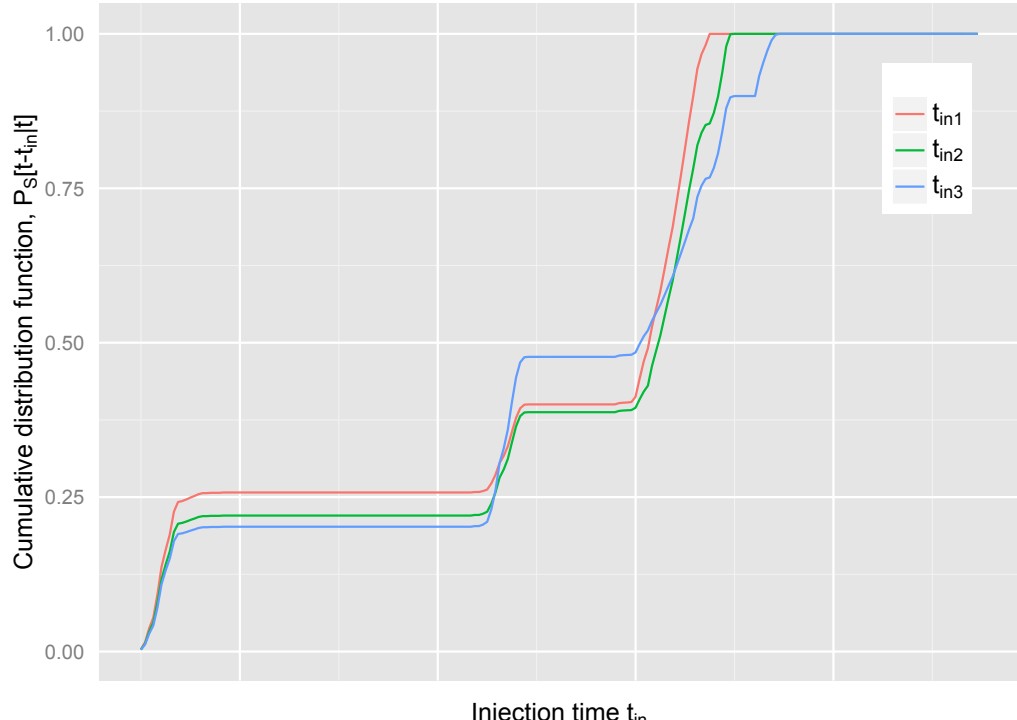

**Figure 3.** Representation of the backward cumulative distribution function for three injection times ($t_{in_i}$, where i = 1,3), as varying with the actual time $t$. The time shift between the three injections was dropped for a direct comparison of the curves.

and

$$V_{AE_T}(t, t_{in}) = \int_0^t ae_T(t, t_{in})dt \qquad (25)$$

$P_S[t - t_{in}|t_{in}]$, as shown in figure 5, varies (with $t$), as expected, between 0 and 1 and has density:

$$p_S(t - t_{in}|t_{in}) = -\frac{1}{J(t_{in})}\frac{ds(t, t_{in})}{dt} = \frac{q(t, t_{in})}{J(t_{in})} + \frac{ae_T(t, t_{in})}{J(t_{in})} \qquad (26)$$

It can be observed instead that:

$$\mathcal{F}(t - t_{in}|t_{in}) := \frac{V_Q(t, t_{in})}{J(t_{in})} \qquad (27)$$

and

$$\mathcal{G}(t - t_{in}|t_{in}) := \frac{V_{E_T}(t, t_{in})}{J(t_{in})} \qquad (28)$$

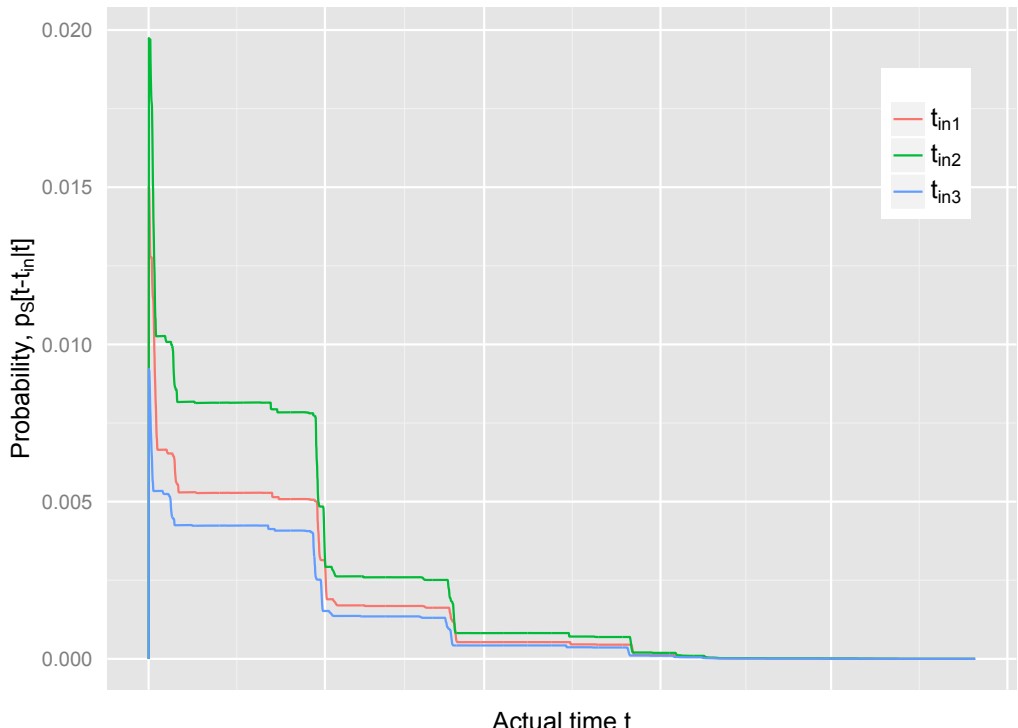

**Figure 4.** Representation of the evolution of the backward pdf versus the actual time $t$. The time shift between the three injections was dropped for a direct comparison of the curves. In this case, the area below the curves is not equal to 1.

are not probability functions, because, their asymptotic value is not 1. Because the forward probabilities are derived, in the case we are describing, on empirical bases from the budgets terms, and not assumed apriori, their complete shape is known only at $t \to \infty$. For any finite time, the actual knowledge we have, is better represented in Figure 6, which shows that the progress of the three curves $P$, $\mathcal{F}$ and $\mathcal{G}$ is unknown for future times.

In order to normalize $\mathcal{F}$ and $\mathcal{G}$, the asymptotic value of the partitioning coefficient is defined among the $Q$ and $E_T$:

$$\Theta(t_{in}) := \lim_{t\to\infty} \Theta(t, t_{in}) := \lim_{t\to\infty} \frac{V_Q(t, t_{in})}{V_Q(t, t_{in}) + V_{E_T}(t, t_{in})} \tag{29}$$

Then, it is easy to show that:

$$p_Q(t - t_{in}|t_{in}) := \frac{q(t, t_{in})}{\Theta(t_{in})J(t_{in})} \tag{30}$$

and

$$p_{E_T}(t - t_{in}|t_{in}) := \frac{ae_T(t, t_{in})}{(1 - \Theta(t_{in}))J(t_{in})} \tag{31}$$

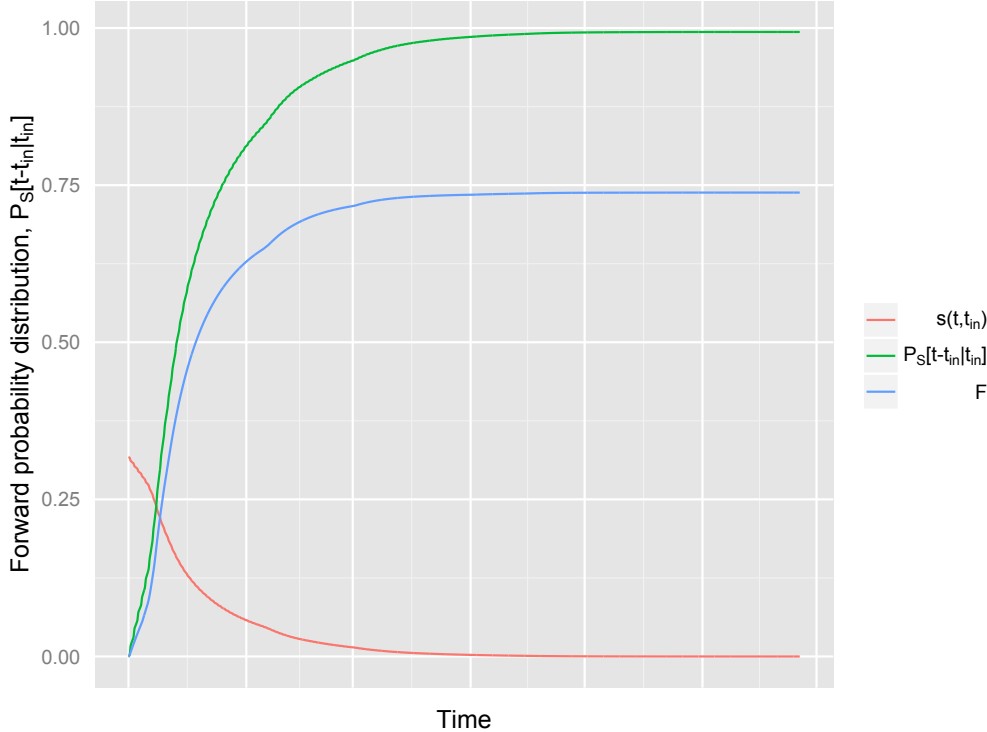

**Figure 5.** Forward residence time probability distribution: in red the relative storage, in green the forward residence time distribution and in blue the relative discharge function.

are the forward probabilities density function of discharges and evapotranspiration, which properly normalize to 1 when integrated over $t$. The two probability density functions $p_Q$ and $p_{E_T}$ are related through:

$$p_S(t - t_{in}|t_{in}) = \Theta p_Q(t - t_{in}|t_{in}) + (1 - \Theta)p_{E_T}(t - t_{in}|t_{in}) \tag{32}$$

Unlike backward probabilities, the forward probabilities describe how a catchment reacts to precipitation events, but they do not describe the actual time the water takes to move through the catchment. To avoid confusion, the expected value of travel time, weighted by the forward distribution, will be called the mean response time (instead of mean travel time). For discharge, the result is:

$$Q(t) = \int_0^{\min(t, t_p)} p_Q(t - t_{in}|t_{in})\Theta(t_{in})J(t_{in})dt_{in} \tag{33}$$

which can be seen as a generalization of the instantaneous unit hydrograph (IUH).

Although $\Theta$ may be unknown at any finite time, the actual state of the system is obtained by solving the budget equation. More information and details on this partitioning coefficient are provided in the next section.

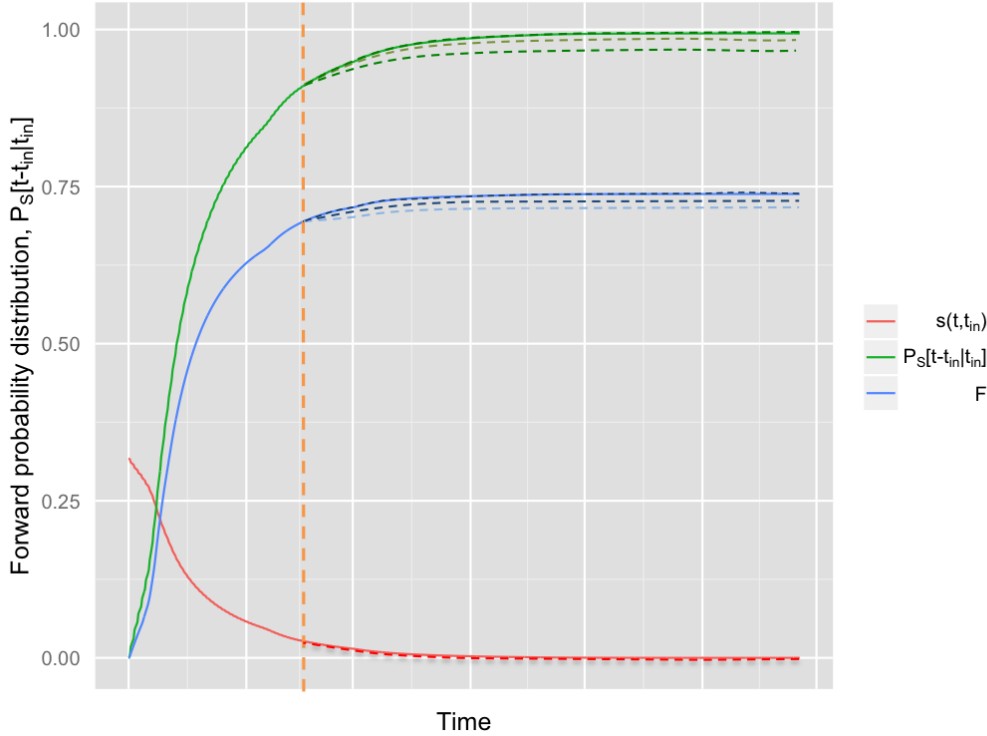

**Figure 6.** Representation of the forward probability of the outputs: in red the relative storage, $s(t, t_{in})$, in green the output probability, $P_S[t - t_{in}|t_{in}]$ and in blue the relative discharge function $\mathcal{F}$, defined in the text. The difference between $P_S[t - t_{in}|t_{in}]$ and $\mathcal{F}$ is the function $\mathcal{G}$, defined in the text. The orange dashed line represents the generic instant $t$, after which $P_S[t - t_{in}|t_{in}]$ and $\mathcal{F}$ are unknown.

## 6 The partitioning coefficient $\Theta$

$\Theta(t_{in})$ has been introduced to complete the algebra of probabilities, in presence of more than one outflow. However estimation of the coefficient is important by itself, because it summarizes the relevant partitioning of hydrologic fluxes.

    The first plot in figure 7 shows a time-series of $\Theta(t, t_{in})$ values obtained from a single injection time considering the complete mixing case ($\omega_Q(t, t_{in}) = \omega_{E_T}(t, t_{in}) = 1$). Data used for the simu-

lation are from Posina River, a small catchment in the North Eastern part of pre-alpine mountainous in Veneto region, Italy. At the beginning $\Theta(t_{in})$ (figure 7, top) shows large oscillations due to hourly and daily oscillations, especially in evapotranspiration. Because $\Theta(t_{in})$ is defined through integrals, these oscillation are progressively damped and become irrelevant after a couple of weeks (when discharge is still higher than baseflow, as appears from the age-ranked disharge in figure 7, bottom).

Figure 8 shows different time-series of the partitioning coefficient: each curve represents the time evolution of $\Theta(t, t_{in})$ obtained considering twelve precipitation events, one for each month of a year

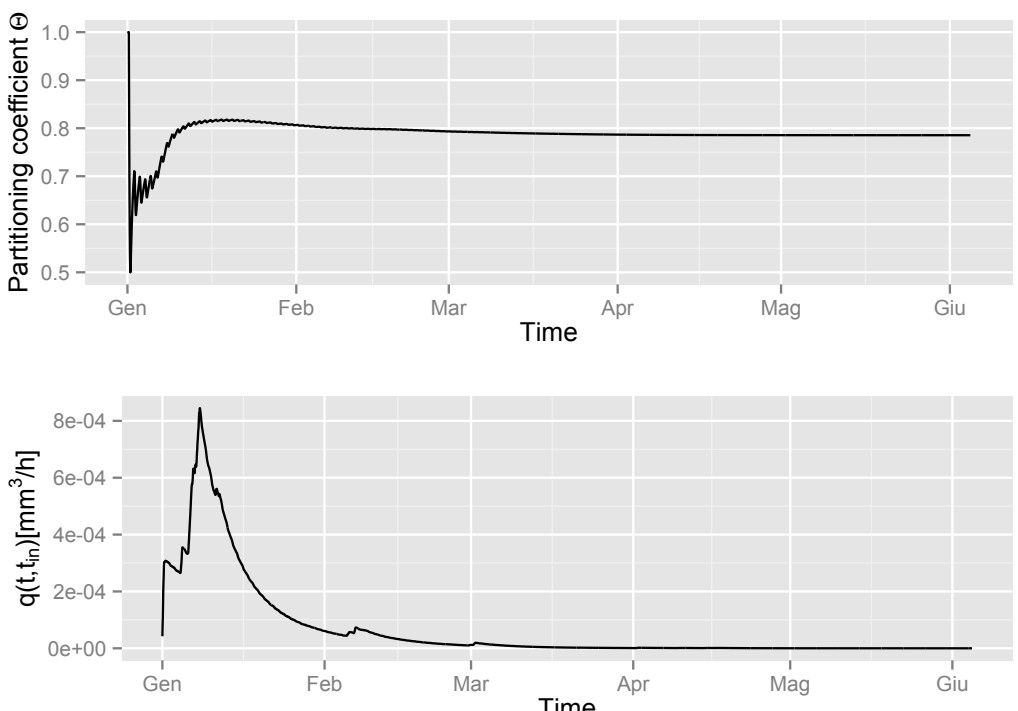

**Figure 7.** Variation of the partitioning coefficient in time, for a single injection time in January: after a time scale of 5 months its oscillation became irrelevant and its value tends to its final value of 0.78

of rainfall data and the considering the complete mixing case ($\omega_Q(t,t_{in}) = \omega_{E_T}(t,t_{in}) = 1$). The highest values of the coefficient ($\Theta(t_{in}) = 0.75$, in this case, are achieved during the coldest months of the year, in which the evapotranspiration flux is lower. On the contrary, smaller $\Theta(t_{in})$ values were obtained in the summer months, with a minimum in June around 0.25.

## 7   Niemi's relation

As a result of definitions made in sections (4) and (5) two relations exist involving $q(t,t_{in})$, i.e. equations (11) and (30), and $ae_T(t,t_{in})$, i.e. equations (12) and (31). Equating the corresponding two expression, results in:

$$Q(t)p_Q(t - t_{in}|t) = \Theta(t_{in})p_Q(t - t_{in}|t_{in})J(t_{in}) \tag{34}$$

and:

$$AE_T(t)p_{E_T}(t - t_{in}|t) = [1 - \Theta(t_{in})]p_{E_T}(t - t_{in}|t_{in})J(t_{in}) \tag{35}$$

where $t = t_{ex}$ since we are considering the particles leaving the control volume as discharge and evapotranspiration. The above relations are known in literature as Niemi's relations or formulas, after Niemi (1977).

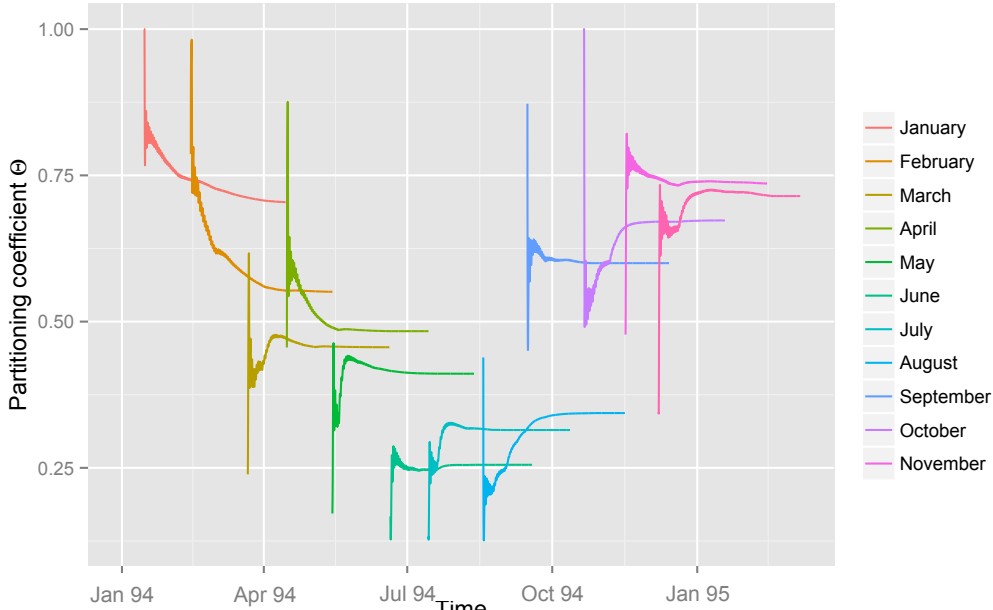

**Figure 8.** Evolution of the partitioning coefficient in one year of hourly simulation: the highest value are achieved in January while the lowest in June. However, the figure does not represent a simple oscillation. March coefficient is lower than April. October abd Novembre present almost the same value.

Defining the total volume of water injected in the system in $[0, t_p]$:

$$V_S(t_p) := \int_0^{\min(t,t_p)} J(t_{in}) dt_{in} = \int_0^{\min(t,t_p)} (Q(t) + AE_T(t)) dt \tag{36}$$

it can be observed that:

$$p_J(t_{in}) := \frac{J(t_{in})}{V_S(t_p)} \tag{37}$$

can be considered the marginal pdf of the injection times, or the fraction of precipitation at a certain discrete $t_{in}$ with respect to the total precipitation over a period of $[0, t_p]$. Analogously

$$p_Q(t) := \frac{Q(t)}{\Theta(t_{in}) V_S(t_{in})} \tag{38}$$

is the marginal pdf of the outflow as discharge, or the fraction of discharge at a certain $t$ generated by precipitation in the same $[0, t_p]$. Then, Niemi's relation (34) becomes:

$$p_Q(t - t_{in}|t) p_Q(t) = p_Q(t - t_{in}|t_{in}) p_J(t_{in}) \tag{39}$$

which has the form of the well known Bayes theorem. This shows that the interpretation of the backward and forward probabilities as conditional ones is fully consistent. On the other hands, this

reveals that the joint probability of $T_r$ and $t$ is:

$$p_S(T_r, t) = p_Q(t - t_{in}|t)p_Q(t) = p_Q(t - t_{in}|t_{in})p_J(t_{in}) \tag{40}$$

Because future in unknown, as remarked in section 5, there should be a working Niemi's relation for any finite time $t$, which does not require the knowledge of the asymptotic value $\Theta(t_{in})$. This can be easily derived after having defined:

$$g(t - t_{in}|t_{in}) := \frac{ae_t(t, t_{in})}{J(t_{in})} \equiv \frac{d\mathcal{G}}{dt} \tag{41}$$

and

$$f(t - t_{in}|t_{in}) := \frac{q(t, t_{in})}{J(t_{in})} \equiv \frac{d\mathcal{F}}{dt} \tag{42}$$

From these definitions,

$$q(t, t_{in}) = f(t - t_{in}|t_{in})J(t_{in}) \tag{43}$$

and

$$ae_t(t, t_{in}) = g(t - t_{in}|t_{in})J(t_{in}) \tag{44}$$

and, therefore,

$$Q(t)p_Q(t - t_{in}|t) = f(t - t_{in}|t_{in})J(t_{in}) \tag{45}$$

for discharges, and

$$AE_T(t)p_{AE_T}(t - t_{in}|t) = g(t - t_{in}|t_{in})J(t_{in}) \tag{46}$$

for evapotranspiration.

As a byproduct, the SAS and the forward functions are shown to be related. For discharge at any time $t$, for example,

$$f(t - t_{in}|t_{in}) = \frac{Q(t)\omega_q(t, t_{in})p_S(t - t_{in}|t)}{J(t_{in})} \tag{47}$$

## 8    Residence times, travel times and life expectancy

The forward probabilities can be related with the life expectancy, i.e. the expected time the water molecules remain in the storage.

In the control volume, we can conceptually denote the subsets of the storage which contains the water molecules expected to exit at time $t_{ex}$ as:

$$s_{t_{ex}}(t, t_{ex}) \tag{48}$$

Analogously to what was done before, we can observe that the quantity

$$300 \quad p_S(t_{ex} - t|t) := \frac{s_{t_{ex}}(t, t_{ex})}{S(t)} \tag{49}$$

has the structure of a probability density function once integrated over all $t_{ex}$-s, and it is reasonable to call it the probability density of storage-life expectancy for particles in the control volume at time $t$.

However, $p_S(t_{ex} - t|t)$ can also be related to the forward probabilities discussed in the previous section. In fact, it can be observed that the probability of storage-life expectancy satisfies the following relation with the age-ranked forward quantities:

$$s_{t_{ex}}(t, t_{ex}) = \int_0^{\min(t,t_p)} \left[ q(t_{ex}, t_{in}) + ae_t(t_{ex}, t_{in}) \right] dt_{in} - \int_0^{\min(t,t_p)} \left[ q(t, t_{in}) + ae_t(t, t_{in}) \right] dt_{in} \tag{50}$$

where, according to the definitions:

$$\int_0^{\min(t,t_p)} \left[ q(t_k, t_{in}) + ae_t(t_k, t_{in}) \right] dt_{in} =$$

$$\int_0^{\min(t,t_p)} \left[ \Theta(t_{in}) p_Q(t_k - t_{in}|t_{in}) + (1 - \Theta(t_{in})) p_{AE_t}(t_k - t_{in}|t_{in}) \right] J(t_{in}) dt_{in} \tag{51}$$

The variable $t_k$, used to make the equations above and below more concise, is such that $t_0 = t_{ex}$ ($k = 0$) and $t_1 = t$ ($k = 1$). The integral spans the time interval up to $t_p$ because we are considering the storage derived for precipitation in the finite interval $[0, t_p]$. In (50) the equality says that the life-storage at time $t$ is equal to the water injected for any time time $t_{in} \in [0, t_p]$ which is expected to exit as discharge or evapotranspiration at time $t_{ex}$. The water still inside the control volume at clock time $t$ is, however, all the water that entered the volume up to time $t$, minus the water that already flowed out.

This integral is not effectively known at time $t$, because what is happening between time $t$ and $t_{ex}$ is unknown, and so the pdfs (as in Figure 5), unless they are specified from some educated guess, as made in the last section of this paper. It follows:

$$p_S(t_{ex} - t|t) = \frac{\sum_{k=0}^1 (-1)^k \int_0^{\min(t,t_p)} \left[ \Theta(t_{in}) p_Q(t_k - t_{in}|t_{in}) + (1 - \Theta(t_{in})) p_{AE_t}(t_k - t_{in}|t_{in}) \right] J(t_{in}) dt_{in}}{S(t)}$$

$$\tag{52}$$

Thus, the relation between the storage-life expectancy and the previously introduced backward and forward probabilities is mediated by an integral equation.

## 9   Passive and reactive solutes

The formalism developed in sections 2 to 6 applies in principle to any conservative substance, indicated by a superscript $i$. Therefore we have a bulk budget equation for the mass of the substance $i$,

and age-ranked budget for the same substance:

$$\frac{dS^i(t)}{dt} = J^i(t) - Q^i(t) + R^i(S(t)) \tag{53}$$

and

$$\frac{ds^i(t,t_{in})}{dt} = j^i(t,t_{in}) - q^i(t,t_{in}) + r^i(s(t-t_{in})) \tag{54}$$

which represent trivial extensions of equations (2) and (9). To simplify this illustration, we have ne-
glected evapotranspiration, which will be re-introduced eventually, but we have added a sink/source
term including any physical or chemical reactions, extending Duffy (2010). However, if the sub-
stance is dissolved in water, it is usually treated as concentration (either in terms of mass, moles or
volume per the same quantity of water). Because we have various terms in the equations, concentra-

tions are possibly as many as the terms that appear. In this case, three:

$$C_S^i(t) := \frac{S^i(t)}{S(t)} \tag{55}$$

for the concentration in storage;

$$C_J^i(t) := \frac{J^i(t)}{J(t)} \tag{56}$$

for concentration in input; and

$$C_Q^i(t) := \frac{Q^i(t)}{Q(t)} \tag{57}$$

for discharges. The latter is actually the one which is usually covered in the literature, since it is
the one measured at the outlet of a control volume/catchment. For the solute discharge, an integral
expression like,

$$Q^i(t) = \int\limits_0^{\min(t,t_p)} \Theta(t_{in}) p_Q(t-t_{in}|t_{in}) J^i(t_{in}) dt_{in} \tag{58}$$

is assumed to be valid, where the $i$ has been dropped from the probability distribution function,
assuming that a passive solute moves with the water. Dividing (58) by the water discharge, it is
obtained:

$$C_Q^i(t) = \int\limits_0^{\min(t,t_p)} \frac{\Theta(t_{in}) p_Q(t-t_{in}|t_{in})}{Q(t)} J^i(t_{in}) dt_{in} \tag{59}$$

and, finally, applying the Niemi's formula:

$$C_Q^i(t) = \int\limits_0^{\min(t,t_p)} p_Q(t-t_{in}|t) \frac{J^i(t_{in})}{J(t_{in})} dt_{in} = \int\limits_0^{\min(t,t_p)} p_Q(t-t_{in}|t) C_J^i(t_{in}) dt_{in} \tag{60}$$

Therefore the concentration of the passive solute in discharge is known once the concentration of the solute in input is known together with the backward probability (Rinaldo et al.,2011). The concentration estimated in this way groups substances injected at any time, in agreement with measurement practices. When a sample is taken, the action implies perfect mixing of all the age-ranked waters in the volume where measurements are made.

The bulk substance budget can instead be written as:

$$\frac{dS^i(t)}{dt} = \frac{dC_S^i(t)S(t)}{dt} = J^i(t) - Q^i(t) + R^i(S(t)) = J^i(t) - C_Q^i(t)Q(t) + R^i(S(t)) \tag{61}$$

and the missing concentration $C_S^i(t)$ can be easily estimated with the help of (55) since $S(t)$ is also known.

The above is essentially the same of equation (12) in Duffy (2010), but the age-ranked formalism can be used to understand a little more about the processes in action. Starting from the quantities that appear in equation (54), the backward probability can be defined as:

$$p^i(t - t_{in}|t) := \frac{s^i(t, t_{in})}{S^i(t)} \tag{62}$$

and analogous definitions (e.g. equation 11) can be given for the discharge and the inputs, such as to obtain, after the appropriate substitutions:

$$\frac{d}{dt}C_S^i(t)S(t)p_S(t - t_{in}|t) = J^i(t)\delta(t - t_{in}) - C_q^i(t)Q(t)\underbrace{\omega_Q(t, t_{in})p_S(t - t_{in}|t)}_{p_Q(t - t_i|t)} + r^i(t, t_{in}) \tag{63}$$

which is the master equation (equation 17) for the substance $i$. Many of the superscripts $i$ were dropped, because the $i$-substance does not modify the velocity (i.e., it behaves like water).

The braces were added to emphasize that $p_Q(t - t_{in}|t)$ should have been left, and we could solve the system of equations directly for $p_S(t - t_{in}|t)$ and $p_Q(t - t_{in}|t)$, obtaining eventually the age-ranked quantities, using (53).

In fact, in (63) all the quantities are known, either because solution of the solute budget (53) or the water master equation (equation 17), or a known input ($J(t)$). The only quantity that is unknown (and usually guessed) is $\omega_Q(t, t_{in})$. However, (63) and (17) can be seen as two coupled equations in $p_S(t - t_{in}|t)$ and $\omega_Q(t, t_{in})$, and we can conclude that the SAS can be derived rather than imposed.

From a practical point of view there could be some obstacles in the correct determination of the SAS, because the distribution of the input of the substance can be unknown. In this case (63) can be used to back-trace the the passive solute injection, after having made educated guesses on the SAS. In the presence of more than one solute, the flow of every solute obeys the same probabilities $p_S$ and $p_Q$. This redundancy can then be used for improving their estimation by applying the appropriate statistical techniques.

For the sake of simplicity we neglected evapotranspiration. However, now that the concepts are established, we can observe that incorporating $AE_T$ involves a second SAS, which remains undetermined. Various approaches can be chosen to overcome this fact. For instance, it can be assumed that

$\omega_Q(t, t_{in}) = \omega_{E_T}(t, t_{in})$. Nevertheless the main experimental approach would be to find a second passive tracer transported through vegetation. In this case, if a third equation similar to (63), but containing evapotranspiration, would hold, it would permit the determination of the missing SAS coefficient.

    Duffy (2010), as in Carrera and Medina (1999), added an equation for water age similar to ours
(53) and (63). This is necessary when dealing with spatially distributed properties (see Appendix B) but not at our spatially integrated scales. In fact, in our case, water age can be estimated directly from its definition (13), since the probability distribution of residence time is known.

    Finally, in order to clarify this theory, an example of $r^i$ could be:

$$r^i(t, t_{in}) := k_1(s^i(t, t_{in}) - k_2 s_{eq}^i) \tag{64}$$

where $k_1$ and $k_2$ are suitable reaction's constants and $s_{eq}^i$ represents an equilibrium storage. Whilst more complex reactions can be envisioned, this type of reaction (or sink term), being linear, does not alter the essential traits of the theory described above.

## 10   A simple example where probabilities are assigned instead than derived.

With the scope to further clarify the formalism, we assume in this section that the forward pdfs
introduced in the previous sections are known. We use the concept of linear reservoir, which has a long history in surface hydrology, e.g. Dooge (2003).

    First consider only one outflow, the bulk equation for the water budget of a single linear reservoir is:

$$\frac{dS(t)}{dt} = \sum_{t_{in}=1}^{n} R_{t_{in}} - \frac{1}{\lambda} S(t) \tag{65}$$

where it has been assumed, for simplicity, that $J(t) = \sum_{t_{in}=1}^{n} R_{t_{in}}$, i.e. that the precipitation is accounted as a sequence of instantaneous impulses at different times $t_{in}$s. By definition of the linear reservoir:

$$Q(t) = \frac{1}{\lambda} S(t) \tag{66}$$

where $\lambda$ [T] is the mean response time (not to be confused with the mean "travel time" derived from
410 the backward distributions) in the reservoir. If this is the case, assuming that the age-ranked storages behave linearly, the age-ranked water budgets can be written as:

$$\frac{ds(t, t_{in})}{dt} = R_{t_{in}} \delta(t - t_{in}) - \frac{1}{\lambda} s(t, t_{in}) \tag{67}$$

where it is

$$q(t, t_{in}) = \frac{1}{\lambda} s(t, t_{in}) \tag{68}$$

Equation (67), after integration over $t_{in}$ reduces to equation (65). By definition, it is $s(t, t_{in}) = 0$ for $t < t_{in}$ and the solution, for $t > t_{in}$ is well known as:

$$s(t, t_{in}) = R_{t_{in}} e^{\frac{t_{in} - t}{\lambda}} \tag{69}$$

The equivalent solution, for $S(t)$ gives:

$$S(t) = \int_{t_{in}}^{t} R_{t_{in}} e^{-(t - t_{in})/\lambda} dt_{in} \tag{70}$$

and the backward probability can be written, then as:

$$p_S(t - t_{in}|t) = \frac{R_{t_{in}} e^{\frac{t - t_{in}}{\lambda}}}{\int_{t_{in}}^{t} R_{t_{in}} e^{-(t - t_{in})/\lambda} dt_{in}} \tag{71}$$

If, and only if, $R_{t_{in}} = const$ the probability simplifies, and it is time invariant, i.e. dependent only on the residence time $T_r = t - t_{in}$. Please notice that, in this case, we did not appeal to equation (17) to estimate the backward probability. Instead we used the definitions in equation (71).

Because discharges are just linearly proportional to the storage, it is easy to show that $p_q(t - t_{in}|t) = p_S(t - t_{in}|t)$ and, therefore, in this case, $\omega(t, t_{in}) = 1$. This shows that the linear reservoir case, where for all injection times the mean residence time is equal (to $\lambda$), the SAS function is necessarily unitary. However, a more general case, can be set if the mean residence time is a function of $t_{in}$, meaning that equation (67) can be modified into:

$$\frac{ds(t, t_{in})}{dt} = R_{t_{in}} \delta(t - t_{in}) - \frac{1}{\lambda_{t_{in}}} s(t, t_{in}) \tag{72}$$

and its solution for $t > t_{in}$ is the same as (69), but with $\lambda$ muted into $\lambda_{t_{in}}$. However, due to the dependence of $\lambda_{t_{in}}$ on the injection time, the SAS is not anymore a constant, being equal to:

$$\omega_Q(t, t_{in}) := \frac{p_q(t - t_{in}|t)}{p_S(t - t_{in}|t)} = \lambda_{t_{in}}^{-1} \frac{\int_{t_{in}}^{t} R_{t_{in}} e^{-(t - t_{in})/\lambda_{t_{in}}} dt_{in}}{\int_{t_{in}}^{t} \lambda_{t_{in}}^{-1} R_{t_{in}} e^{-(t - t_{in})/\lambda_{t_{in}}} dt_{in}} = \lambda_{t_{in}}^{-1} \frac{\int_{t_{in}}^{t} R_{t_{in}} e^{+t_{in}/\lambda_{t_{in}}} dt_{in}}{\int_{t_{in}}^{t} \lambda_{t_{in}}^{-1} R_{t_{in}} e^{t_{in}/\lambda_{t_{in}}} dt_{in}} \tag{73}$$

This seems to suggest that imposing the characteristics of the pdf could completely determine the $\omega_Q(t, t_{in})$. Vice versa, as already known, assigning $\omega_Q(t, t_{in})$ from some heuristic, obviously, would determine a mean residence time dependence on the injection time.

Non trivial $\omega(t, t_{in})$ can also be derived from assuming a sequence of linear reservoirs, as in the so called Nash model, Dooge (2003). Without entering in details, a sequence of linear reservoirs implies that just the last reservoir maintains a linear relation between storage and outflow. Instead a nonlinear relationship exists between the whole storage and the same outflow, implying also a nonlinear SAS.

Even if semi-analytical results are not feasible using non-linear reservoirs, suitably tuning the parameters of each age-ranked equation cannot change the form of the SAS , as is also suggested by arguments below.

Other aspects come into play when there are multiple outputs. Expanding the previous linear case to include evapotranspiration, the bulk equation, becomes:

$$\frac{dS(t)}{dt} = \sum_{t_{in}=1}^{n} R_{t_{in}} - \left(\frac{1}{\lambda} - aet(t)\right) S(t) \tag{74}$$

where the actual evapotranspiration is assumed to equal:

$$AE_T(t) = S(t)aet(t) \tag{75}$$

with a linear dependence on the soil water content, as for instance in Rodriguez-Iturbe et al. (1999). The equations of water budget for the generations becomes:

$$\frac{ds(t,t_{in})}{dt} = R_{t_{in}}\delta(t - t_{in}) - \left(\frac{1}{\lambda_{t_{in}}} + ae(t,t_{in})\right) s(t,t_{in}) \tag{76}$$

where the bivariate dependence of $ae(t,t_{in})$ on the actual time and the injection time can be justified by arguing that, water of different ages is not perfectly mixed in the control volume and plant roots
sample water of different ages in different modes, according to their spatial distributions. Since equation (76) remains a linear ordinary differential equation, it can be solved analytically, and:

$$s(t,t_{in}) = R_{t_{in}} e^{-\Lambda(t,t_{in})} \tag{77}$$

where:

$$\Lambda(t,t_{in}) := \int_{t_{in}}^{t} \left(\frac{1}{\lambda_{t_{in}}} + ae(t',t_{in})\right) dt' \tag{78}$$

and:

$$S(t) = \int_{0}^{t} R_{t_{in}} e^{-\Lambda(t,t_{in})} dt_{in} \tag{79}$$

Notably, the outflows terms can be expressed as a function of the storage:

$$q(t,t_{in}) + aet(t,t_{in}) = \mu(t,t_{in}) s(t,t_{in}) \tag{80}$$

the problem remains linear and analytically solvable. The quantity $\mu(t,t_{in})$ is usually called age
and mass-specific output rate, Calabrese and Porporato (2015). Solving equation (76) it is not even necessary to show that:

$$\omega_{E_T}(t,t_{in}) \neq 1 \tag{81}$$

The latter condition is regained if and only if $aet(t,t_{in}) = aet(t)$, i.e. it depends only on the current time (which is a condition that requires the perfect mixing of aged waters). In fact, in case a dependence on $t_{in}$ remains, then, trivial algebra says that:

$$p_{E_T}(t - t_{in}|t) = \frac{ae(t,t_{in})s(t,t_{in})}{\int_{t_{in}}^t ae(t,t_{in})s(t,t_{in})dt_{in}}$$

(82)

which implies:

$$\omega_{E_T}(t,t_{in}) := \frac{p_{E_T}(t - t_{in}|t)}{p_S(t - t_{in}|t)} = \frac{ae(t,t_{in})\int_{t_{in}}^t R_{t_{in}} e^{-\Lambda(t,t_{in})}}{\int_{t_{in}}^t ae(t,t_{in})S(t,t_{in})dt_{in}}$$

(83)

Obviously these results, obtained by imposing a travel time probability, can be inconsistent with tracers results, because both approaches require estimates of the $\omega$ functions, which are not known well.

## 11    Conclusions

We reviewed existing concepts that were collected from many different papers, and presented them in a new systematic way. We established a consistent framework that offers a unified view of the travel time theories across surface water and groundwater. It contains several clarifications and extensions.

Clarifications include:

– the concepts of forward and backward conditional probabilities and a small but important change in notation;

– their one-to-one relation with the water budget (and the age-ranked functions) from which the probabilities were derived (after the choice of SASs);

– the proper way to choose backward probabilities. Specifically, it was shown that the usual way to assign time invariant backward probabilities is inappropriate. We also show how to do it correctly, and introduced a minimal time variability.

– the fact that time-invariant forward probabilities usually imply time-varying backward probabilities, i.e. travel time distributions.

– the rewriting of the master equation by Botter, Bertuzzo and Rinaldo as an ordinary differential equation (instead of a partial differential equation);

– the role and nature of the partitioning coefficient between discharge and evapotranspiration (which is unknown at any time except asymptotically);

– the significance of the SAS functions with examples;

- the relationship of the present theory with the well known theory of the instantaneous unit hydrograph; and

500 - we added information and clarified some links of the present theory with [Delhez et al. (1999) and [Duffy (2010)].

Extensions include:

- new relations among the probabilities (including the relation between expectancy of life and forward residence time probabilities);

- an analysis of the partitioning coefficients (which are shown to vary seasonally);

- an explicit formulation of the equations for solutes which would permit direct determination of the SAS on the basis of experimental data;

- tests of the effects of various hypotheses, e.g., assuming a linear model of forward probability and gamma model for the backward probabilities;

- an extension of Niemi's relation (and a new normalization);

- the presentation of Niemi's relation as a special case of the Bayes Theorem; adn

- a system of equations from which to obtain the SAS experimentally.

The extension of the theory to any passive substance diluted in water clearly opens the way to new developments of the theory and applications of tracers.

Finally, as a proof of concept, this paper includes examples derived from a real case (Posina River 515 Basin) and comes with open source code that implements the theory, available to any researcher (see Appendix E).

## Appendix A: Symbols, Acronyms, and Notation

| Symbol | Name | Units |
|--------|------|-------|
| $ae_T(t, t_{in})$ | age-ranked evapotranspiration | $L^3 T^{-2}$ |
| $ae_T(t, t_{ex})$ | age-ranked evapotranspiration conditioned to the exit time | $L^3 T^{-2}$ |
| $b$ | exponent of the non-linear reservoir model | $-$ |
| $f(t - t_{in}\|t_{in})$ | time derivative of the relative discharge function | $T^{-1}$ |
| $f_{up}$ | partitioning coefficient between upper and saturated reservoirs | $-$ |
| $g(t - t_{in}\|t_{in})$ | time derivative of the relative evapotranspiration function | $T^{-1}$ |
| $g(T_r)$ | incomplete Gamma distribution | $T^{-1}$ |
| $j(t, t_{in})$ | age-ranked rainfall rate | $L^3 T^{-2}$ |
| $j^i(t, t_{in})$ | age-ranked input of the substance $i$ | $L^3 T^{-2}$ |
| $k_{1,2}$ | reaction's constants | $-$ |
| $p^i(t - t_{in}\|t)$ | travel time backward pdf of the substance $i$ | $T^{-1}$ |
| $p_{E_T}(t - t_{in}\|t)$ | evapotranspiration time backward pdf | $T^{-1}$ |
| $p_{E_T}(t - t_{in}\|t_{in})$ | evapotranspiration time forward pdf | $T^{-1}$ |
| $p_J(t_{in})$ | marginal pdf of the outflow as discharge | $-$ |
| $p_{low}(t - t_{in}\|t)$ | travel time backward pdf of the lower storage | $T^{-1}$ |
| $p_Q(t - t_{in}\|t)$ | travel time backward pdf | $T^{-1}$ |
| $p_Q(t - t_{in}\|t_{in})$ | travel time forward pdf | $T^{-1}$ |
| $p_Q(t_{in})$ | marginal pdf of the injection times | $-$ |
| $p_S(T_r\|t)$ | residence time backward pdf | $T^{-1}$ |
| $p_S(t - t_{in}\|t_{in})$ | residence time forward pdf | $T^{-1}$ |
| $p_S(t_{ex} - t\|t)$ | life expectancy backward pdf | $T^{-1}$ |
| $p_{sat}(t - t_{in}\|t)$ | travel time backward pdf of the saturated storage | $T^{-1}$ |
| $p_{sup}(t - t_{in}\|t)$ | travel time backward pdf of the upper storage | $T^{-1}$ |
| $q(t, t_{in})$ | age-ranked discharge | $L^3 T^{-2}$ |
| $q(t, t_{ex})$ | age-ranked discharge conditioned to the exit time | $L^3 T^{-2}$ |
| $q^i(t, t_{in})$ | age-ranked output of the substance $i$ | $L^3 T^{-2}$ |
| $q_{low}(t, t_{in})$ | age-ranked discharge for the lower reservoir | $L^3 T^{-2}$ |
| $q_{sat}(t, t_{in})$ | age-ranked discharge for the saturated reservoir | $L^3 T^{-2}$ |
| $r^i(t, t_{in})$ | age-ranked sink/source term | $L^3 T^{-2}$ |
| $s(t, t_{in})$ | age-ranked water storage | $L^3 T^{-1}$ |
| $s^i(t, t_{in})$ | age-ranked water storage of the substance $i$ | $L^3 T^{-2}$ |
| $s^i_{eq}$ | equilibrium storage | $L^3 T^{-1}$ |

| Symbol | Name | Units |
|---|---|---|
| $s_{ex}(t, t_{ex})$ | age-ranked water storage conditioned to the exit time | $L^3 T^{-1}$ |
| $s_{low}(t, t_{in})$ | age-ranked water storage for the lower reservoir | $L^3 T^{-1}$ |
| $s_{up}(t, t_{in})$ | age-ranked water storage for the upper reservoir | $L^3 T^{-1}$ |
| $s_{sat}(t, t_{in})$ | age-ranked water storage for the saturated reservoir | $L^3 T^{-1}$ |
| $t$ | actual time | $T$ |
| $t_{ex}$ | exit time | $T$ |
| $t_{in}$ | injection time | $T$ |
| $t_p$ | time of the end of the last precipitation considered in the analysis | $T$ |
| $AE_T(t)$ | actual evapotranspiration | $L^3 T^{-1}$ |
| $C_J^i(t)$ | concentration in input | $-$ |
| $C_S^i(t)$ | concentration in storage | $-$ |
| $C_Q^i(t)$ | concentration in discharge | $-$ |
| $E(t)$ | potential evapotranspiration | $L^3 T^{-1}$ |
| $\mathcal{F}(t - t_{in}|t_{in})$ | relative discharge function | $-$ |
| $\mathcal{G}(t - t_{in}|t_{in})$ | relative evapotranspiration function | $-$ |
| $J(t)$ | rainfall rates | $L^3 T^{-1}$ |
| $J^i(t)$ | input rates of the substance $i$ | $L^3 T^{-1}$ |
| $J(t_{in})$ | precipitation at a certain $t_{in}$ | $L^3 T^{-1}$ |
| $P_S(t - t_{in}|t_{in})$ | residence time forward probability function | $-$ |
| $L_e$ | life expectancy | $T$ |
| $T$ | travel time | $T$ |
| $T_r$ | residence time | $T$ |
| $S(t)$ | volume of water stored in a control volume | $L^3$ |
| $Q(t)$ | discharge | $L^3 T^{-1}$ |
| $Q^i(t)$ | output rates of the substance $i$ | $L^3 T^{-1}$ |
| $Q_1$ | recharge to the saturated reservoir | $L^3 T^{-1}$ |
| $Q_l$ | runoff produced by the lower reservoir | $L^3 T^{-1}$ |
| $Q_{sat}$ | outflow from the saturated storage | $L^3 T^{-1}$ |
| $R^i(S(t))$ | sink/source term | $L^3 T^{-1}$ |
| $R(t)$ | recharge to the lower reservoir | $L^3 T^{-1}$ |
| $R(t, t_{in})$ | input to the lower reservoir | $L^3 T^{-1}$ |
| $R_{t_{in}}$ | sequence of instantaneous impulses at different $t_{in}s$ | $L^3$ |

| Symbol | Name | Units |
|---|---|---|
| $S^i(t)$ | stored mass of the substance $i$ stored | $L^3$ |
| $S_{low}$ | storage in the lower reservoir | $L^3$ |
| $S_{max}$ | maximum value of the storage | $L^3$ |
| $S_{sat}$ | amount of water stored in the saturated storage | $L^3$ |
| $S_{up}$ | storage in the upper reservoir | $L^3$ |
| $V_{AE_T}(t,t_{in})$ | time integral of the age-ranked evapotranspiration | $L^3 T^{-1}$ |
| $V_S(t_p)$ | total volume injected in the volume in $[0,t_p]$ | $L^3 T^{-1}$ |
| $V_Q(t,t_{in})$ | time integral of the age-ranked discharge | $L^3 T^{-1}$ |
| $\alpha$ | coefficient of the gamma distribution | — |
| $\delta(t-t_{in})$ | Delta-dirac distribution | $T^{-1}$ |
| $\gamma$ | coefficient of the gamma distribution | — |
| $\lambda$ | coefficient of the non-linear reservoir model | $T$ |
| $\mu(t,t_{in})$ | age and mass-specific output rate | — |
| $\omega_{E_T}(t,t_{in})$ | SAS for evapotranspiration | — |
| $\omega_{low}(t,t_{in})$ | SAS for runoff produced by the lower reservoir | — |
| $\omega_Q(t,t_{in})$ | SAS for discharge | — |
| $\omega_{Q_1}(t,t_{in})$ | SAS for the recharge to the saturated reservoir | — |
| $\omega_R(t,t_{in})$ | SAS for the recharge to the lower reservoir | — |
| $\omega_{Q_{sat}}(t,t_{in})$ | SAS for runoff produced by the saturated storage | — |
| $\Theta(t_{in})$ | partitioning coefficient | — |
| $\Gamma$ | Gamma function | — |

## Appendix B:  A little critical review of contributions on age related equations

Without the need to be comprehensive, since some review of the topic were recently made available, (Benettin et al., 2013; Hrachowitz et al., 2016), we believe it could be useful to summarize the contributions of some milestone papers in relation to our. We choose here those references that have a direct theoretical influence, and leave out those, already cited in the main text, having more relevance in connection with experimental research and model identification. We do not mention

also Dagan's important work that we already commented in Introduction.

We also do not mention travel time theories which emanate from the instantaneous unit hydrograph, since they were extensively discussed in Rigon et al. (2016). The formal center of this paper contribution is equation (9). Being substantially a mass budget, it can be argued that it has been central in many scientific disciplines and hydrology's sub-disciplines. However, as stated in the main

text, (van der Velde et al., 2012) is the first contribution where the equation appears in the form we use.

One of the older papers on this subject is Campana (1987) who wrote an equation for water age distribution, but he used a discrete time formalism that is not easily translatable into our derivation. The remarkable work of Carrera and Medina (1999) was directly paying attention to the question of water ages by finding one partial differential equation (pde) for the residence time distributions, and one pde for water ages. A similar approach was also followed by Ginn (1999). Their contributions fall in the area of advection-dispersion type of equations and were implemented, almost at the same time in Delhez et al. (1999) and Deleersnijder et al. (2001). The latter concerned with the oceanography domain. Parallel developments in atmospheric sciences are instead reviewed in Waugh and Hall (2002). All the researchers above worked at a finer scale than our, describing fields of properties dependent on location, time and age, while we work at a scale integrated over a whole control volume (a catchment or a hydrologic response unit), where any reference to space disappears. Let us call their approach "local" and our approach "spatially integrated". Their local approach used directly concentrations while our spatially integrated put emphasis on residence (and travel) time probabilities. Both concentration and probability vary between zero and one but the first are mass (volume) normalised over the total mass (volume) of all substances present in a given location, the second are mass (volumes) of a substance injected at a certain time over the mass (volume) of the same substance coming from all the injection times. We have shown in section 9 how the two approaches match at the spatially integrated scale following the work of Duffy (2010). Another relevant difference between the local and spatially integrated theories is the different parameterisation of the fluxes. In our treatment we distinguish the sources (precipitation, recharge, etc) and the outputs (discharges and evapotranspiration). Local theories usually implement an advection-dispersion term and include a sink-source term, which is important only when solutes are involved. We also introduced a sink-source term, but when appropriate, in section 9. An explicit integration of the local theory to obtain the spatially integrated one was recently presented in Duffy (2010) who first made clear that the equation for concentration and mass budget form a dynamical system. He also added an age equation which we do not need in our formalism.

Porporato and Calabrese (2015) and Calabrese and Porporato (2015) in their effort to merge the travel time approach with population dynamics, dated the age ranked equation back to the work of M'Kendrick and Foerster (M'Kendrick, 1925; Foerster, 1959, e.g.,). However M'Kendrick and Foerster version of the master equation emphasizes more the birth and death terms (i.e. the sink and sources of the local theories mentioned above), instead of the flows at the interfaces, as it is usually done when dealing with hydrological budgets. Even if this approach is interesting, as Rinaldo et al. (2015) notes, it is very difficult to work out hydrology in terms of the loss function which is, instead, central in the population dynamic. If population dynamics theories could be considered one of our

ancestors, they are not focused directly on the same information. With the same argument can be commented Rotenberg (1972) work.

A different but interleaved group of papers, e.g. Kirchner (2016a, b), and references therein, Hrachowitz et al. (2010), analyses the topic of tracers flow by directly assigning the backward probability, in (60). This approach, as well IUH related ones (shown in the main text), could determine the forward travel time distribution through the Niemi's relation. However, as shown in appendix D, this approach is not respecting the definition of probabilities we gave, and actually has some mathematical inconsistency which should, in future, be corrected.

## Appendix C: An example of generalization to many embedded reservoirs

In the literature we cited in the main text, it seems usually recognized that a single reservoir is not able to reproduce proper discharge and tracers behavior, and a few "embedded" reservoirs are therefore used in models. For instance, concerns regarding the discrepancies between the velocity of the solute transport and celerity of the pressure signals that travel across the control volumes must be addressed with an appropriate choice of embedded "groundwater" reservoirs.

The theory developed in the main text can be easily extended to these cases. As an illustrative example we take a simple model from Birkel et al. (2010) and Soulsby et al. (2015).

Their system is composed by three reservoirs (e.g. Figure 2 in Soulsby et al. (2015)). The lower reservoir is a responsible for groundwater description and represents a large storage which has also the function to dump the solute concentration. The other two reservoirs are at the surface. The first takes precipitation $J$, produces evapotranspiration $ET$, and returns recharge $R$ for the lower reservoir and some outflow that goes into the second reservoir. This second reservoir is assumed to reproduce the behavior of a saturated riparian zone that originates the surface runoff flowing into channels. The budget equations are written below.

$$\frac{dS_{up}(t)}{dt} = (1 - f_{sup})\, J(t) - ET(t) - Q_1(t) - R(t) \tag{C1}$$

where $S_{sup}$ is the amount of water stored in the upper reservoir, $f_{sup}$ is a coefficient that separates the amount of water and evapotranspiration that pertain to the upper storage from those of the saturated reservoir, $Q_1$ is the discharge into the saturated reservoir, and $R$ is the recharge to the groundwater (lower) storage. In this budget equation $f_{sup}$ is a given parameter, and $ET$ is a measured function (but making it a modeled quantity dependent on water storage does not change anything substantial). Both the other outflows are determined as linear functions of the storage $S_{sup}$ as:

$$Q_1(t) = a\, S_{up}(t) \tag{C2}$$

and

$$R(t) = b\, S_{up}(t) \tag{C3}$$

where the two coefficients $a$ and $b$ are assumed to be given, after an appropriate process of calibration. With all of these assumption eq. (C1) is analytically solvable, and $S_{sup}$ can be considered known. Applying the theory developed in the main text, the age-ranked equations for this storage are given by:

$$\frac{dS_{up}(t)p_{sup}(t-t_{in}|t)}{dt} = (1-f_{sup})\,J(t_{in})\delta(t-t_{in})-ET(t)-Q_1(t)\omega_{Q_1}(t,t_{in})p_{sup}(t-t_{in}|t)-\omega_R(t,t_{in})R(t)p_{sup}(t-t_{in}|t)$$

(C4)

Once the two SASs in Eq. (C4), i.e. $\omega_{Q_1}(t,t_{in})$ and $\omega_R(t,t_{in})$, are assigned, also the probability $p(t-t_{in}|t)$, and the age-ranked storage $s(t,t_{in})$ can be determined. As usual, in these cases, the authors assumed $\omega_{Q_1}(t,t_{in}) = \omega_R(t,t_{in}) = 1$.

The lower reservoir obeys the following budget equation:

$$\frac{dS_{low}(t)}{dt} = R(t) - kS_{low}(t)$$

(C5)

where $Q_2 = kS_{low}(t)$ is the runoff produced by seepage, and $k$ is a calibration coefficient. Since $R(t)$ is known from solving the upper reservoir, also Eq. (C5), is solvable. Eq. (C5) can then be associated with the age-ranked master equation:

$$\frac{dS_{low}(t)p_{low}(t-t_{in}|t)}{dt} = R(t,t_{in}) - bS_{low}(t)\omega_{low}(t,t_{in})p_{low}(t-t_{in}|t)$$

(C6)

where $R(t,t_{in})$ is the input to the second reservoir which comes with aged waters, and is given by solving Eq. (C4) because it is $R(t,t_{in}) = R(t)p(t-t_{in}|t)$. In turn Eq. (C6) is solvable and can be used to obtain all the age-ranked functions relative to the lower storage. Notably, all the above four differential equations are linear and therefore analytically solvable as functions of the inputs, even if the analytic solutions are not reported here.

Finally, the storage equation for the saturated storage is:

$$\frac{dS_{sat}(t)}{dt} = f_{sup}(J(t) - ET(t)) + Q_1(t) - Q_{sat}(t)$$

(C7)

where $Q_1$ is the input from the upper reservoir and the outflow to channels is described with a non-linear reservoir law:

$$Q_{sat}(t) = rS_{sat}^{1+\beta}$$

(C8)

and $r$ and $\beta$ are two further coefficients to be calibrated. In total, this system of embedded reservoirs contains five parameters for calibration, $a$, $b$, $k$, $r$ and $\beta$.

Following the same arguments as for the other two reservoirs, the age-ranked version of the budget becomes:

$$\frac{dS_{sat}(t)p_{sat}(t-t_{in}|t)}{dt} = f_{sup}(J(t_{in})\delta(t-t_{in})-ET(t))+Q_1(t,t_{in})-Q_{sat}(t)\omega_{Q_{sat}}(t,t_{in})p_{sat}(t-t_{in}|t)$$

(C9)

As in the case of the lower reservoir, the saturated reservoir receives aged waters from the upper one. The equation is not analytically solvable, but well known numerical methods can produce the solution easily.

The overall system is the sum of the three reservoirs where:

$$S(t) = S_{up}(t) + S_{low}(t) + S_{sat}(t) \qquad \text{(C10)}$$

and

$$s(t, t_{in}) = s_{up}(t, t_{in}) + s_{low}(t, t_{in}) + s_{sat}(t, t_{in}) \qquad \text{(C11)}$$

Therefore

$$p_S(t - t_{in}|t) := \frac{s(t, t_{in})}{S(t)} \qquad \text{(C12)}$$

is the backward residence time distribution for the compound system. Because

$$Q(t) = Q_{low}(t) + Q_{sat}(t), \qquad \text{(C13)}$$

and

$$q(t, t_{in}) = q_{low}(t, t_{in}) + q_{sat}(t, t_{in}), \qquad \text{(C14)}$$

the global travel time distribution is:

$$p_Q(t - t_{in}|t) := \frac{q(t, t_{in})}{Q(t)} \qquad \text{(C15)}$$

It follows that the compound systems behaves like having a SAS given by:

$$\omega(t, t_{in}) = \frac{p_S(t - t_{in}|t)}{p_Q(t - t_{in}|t)} \qquad \text{(C16)}$$

On the basis of the global probability distribution functions, the behavior of a tracer $i$ can be obtained from Niemi's relations as:

$$C_Q^i(t) = \int\limits_0^{\min(t, t_p)} p_Q(t - t_{in}|t) C_J^i(t_{in}) dt_{in} \qquad \text{(C17)}$$

This concentration does not distinguish between waters coming from the saturated and the lower reservoir. However, the theory can do it by substituting Eq. (C17) in place of $p_Q(t - t_{in}|t)$, $p_{Q_{low}}(t - t_{in}|t)$ or $p_{Q_{sat}}(t - t_{in}|t)$. Because it must be:

$$p_Q(t - t_{in}|t) = (1 - \Theta_Q(t)) p_{Q_{low}}(t - t_{in}|t) + \Theta_Q(t) p_{Q_{sat}}(t - t_{in}|t) \qquad \text{(C18)}$$

where:

$$\Theta_Q(t) = \frac{Q_{sat}(t)}{Q_{sat}(t) + Q_{low}(t)} \qquad \text{(C19)}$$

is the appropriate partitioning coefficient. To obtain the last equations, it is sufficient to apply the definitions for the probabilities. The case treated is general enough to show that any set of coupled reservoirs can be analyzed from the travel time point of view, no matter how complex the system is.

## Appendix D:  An observation on fixing the functional form of the backward probability

It can be observed that the backward probability, as defined in (10) is quite restrictive, and not very compatible with the assumption of a time invariant backward distribution, often made in literature, (e.g. Kirchner et al., 2000; Kirchner, 2016a; Hrachowitz et al., 2010). Most of these papers use a gamma distribution, i.e.

$$g(T_r) = \frac{T_r^{\alpha+1} e^{\frac{T_r}{\gamma}}}{\gamma^\alpha \Gamma(\alpha)} \tag{D1}$$

where $g$ is the incomplete gamma distribution, $T_r := t - t_{in}$ is the residence time, $\alpha$ and $\gamma$ are the two coefficient of the incomplete $\Gamma$ distribution, $\Gamma$ is the gamma function. $g(T_r)$ in (D1) is certainly a distribution though over the whole domain of $T_r$. However, equation (10) requires that $g(T_r)$ would be a probability for any clock time $t$, i.e. that:

$$\int\limits_0^{\min(t,t_p)} p_Q(t - t_{in}|t) dt_{in} = 1 \tag{D2}$$

This is, clearly not obtained with $(D1)$ (or any other classical distribution), and, in fact,

$$\int\limits_0^{\min(t,t_p)} \frac{(t-t_{in})^{\alpha+1} e^{\frac{(t-t_{in})}{\gamma}}}{\gamma^\alpha \Gamma(\alpha)} dt_{in} \neq 1 \tag{D3}$$

where in the formula the travel time $T_r$ has been explicitly written as function of $t$ and $t_{in}$. It could be argued that the above integral could be approximately equal to unity in real cases, and, seen the success of gamma based approaches to interpret experimental data, this could be true.  However, a

better choice for the backward probability should be a little more complex. For instance:

$$p_Q(t - t_{in}|t) = \frac{g(t - t_{in})}{\int_0^{\min(t,t_p)} g(t - t_{in}) dt_{in}} = \frac{\frac{(t-t_{in})^{\alpha+1} e^{\frac{(t-t_{in})}{\gamma}}}{\gamma^\alpha \Gamma(\alpha)}}{\int_0^{\min(t,t_p)} \frac{(t-t_{in})^{\alpha+1} e^{\frac{(t-t_{in})}{\gamma}}}{\gamma^\alpha \Gamma(\alpha)} dt_{in}} \tag{D4}$$

works the right way.

## Appendix E:  Reproducible research

For interested researchers to replicate or extend our results, our codes are made available at https:
//github.com/geoframecomponents. Instructions for using the code can be found at: http://geoframe. blogspot.com. All the material, with further information, is also linked at http://abouthydrology. blogspot.com/search/label/Residence%20time.

*Acknowledgements.* The authors acknowledge the Trento University project CLIMAWARE (/http://abouthydrology.blogspot.it/search/label/CLIMAWARE) and the European Union FP7 Collaborative Project

GLOBAQUA (Managing the effects of multiple stressors on aquatic ecosystems under water scarcity, grant no. 603629-ENV-2013.6.2.1) that partially financed this research. They also thank Professor Aldo Fiori to have indicated them some relevant literature on the subject. We thank Dr. Wuletawu Abera for having provided his Posina catchment simulations. Finally we thank dr. Markus Hrachowitz, dr. Paolo Benettin, and mr. Daniel Wilusz for their wotk that helped to enhance our initial manuscript with their reviews.

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
