# Peer review of "Age-ranked hydrological budgets and a travel time description of catchment hydrology"

_Hydrology and Earth System Sciences, 2016_

## Referee Comment (RC1) · Anonymous Referee #1 · 13 Jun 2016

The value of a travel time based description of catchment hydrology has been increasingly acknowledged over the recent years. Therefore, this manuscript comes timely and could eventually be an interesting contribution to literature. Presenting a considerable string of formalisms, describing various aspects of travel times, and which are, as far as I can see, mathematically sound, the authors delve deep into the topic. However, even after reading the manuscript three times I struggle to see what the actual intended contribution is. What do the authors want to convey to the reader? This needs to be made much clearer. Is it a review of existing concepts? Is it an extension of existing concepts? If it is a review, the description of the concept needs to go further back to include earlier work and detailed descriptions thereof. If it is rather an extension of existing concepts, it needs to be clarified what the novelty is and how it fits into our current understanding. In other words, what are the main findings? What do we learn?

[Figure]

In either case, the presented work needs to be put into a broader context. The authors refer to a few key publications, but they ignore many other recent contributions that address the issue from similar and/or different perspectives. Such a broader context will help the reader to better appreciate the relevance of the presented work. I would thus invite the authors to discuss their methods and findings with respect to methods, results and findings from a wider range of other (also more experimental) studies including, for example Birkel et al. (2010, 2011, 2014), Fenicia et al. (2010), Van der Velde et al. (2010, 2015), McMillan et al. (2012), Hrachowitz et al. (2015), Rinaldo et al. (2015) but also with the work of Cvetkovic, Fiori, Dagan, etc over the past years.

Other comments:

(1) P.1, l.17-18: there is, in my understanding, little that remains unclear. Perhaps provide some examples.

(2) P.2, l.26: again, I sort of disagree, there is little that remains unexplained. Please also give an example here to clarify.

(3) P.2, l.33-41: please be more specific here: what is the research hypothesis to be tested?

(4) P.2,l.47 and 48: this should read as "...the time at which..." to avoid confusion, as we are (as a simplifying assumption) not talking about a time interval over which the input occurs but an instantaneous input.

(5) P.3,l.57-58: please add the respective dimensions

(6) P.3,l.67: this can only be solved analytically if piecewise linear functions of inputs are available and, more importantly, with the assumption of only one storage component in the system, which may be quite an oversimplification for most catchments. The water balance as given here cannot resolve the non-linearites in the system, including interception, vadose zone dynmaics, storm flow connectivity, etc.. Thus the practical utility of such an analytical solution, if not used in an operator splitting strategy that accounts for different system components, remains limited. Please qualify the statement accordingly.

(7) P.4,l.86: Really? I would be surprised by that as this is implicit (and has to be) in essentially all approaches that somehow track fluxes through the system. I could imagine that this has already been explicitly formulated earlier. Please check, in particular papers by Cvetkovic, Dagan, Fiori, Russo, etc.!

(8) P.5, section 4: that is all fine and true, but nothing new. It remains unclear what the purpose of this section is. Please clarify!

(9) P.6,l.136: I may have missed something here, but how can P(Tr/t) not integrate to one (and it seems it actually does in figure 2)?? As far as I understand, it is the sum of storages of all given ages present in the system over the total storage at any time t. Also: how does ageing contribute here? Please clarify.

(10) Figures 2 and 4: I am a bit confused by this figure. How can three injections at three different injection times (tau1, tau2, tau3) plot on top of each other when the x-axis is the actual time t measured by a clock? Should here the x-axis not rather be the time since injection?

(11) P.8,l.161ff: that is correct, but has been shown and discussed earlier (e.g. Benettin et al., 2015, Fig.6; Hrachowitz et al., 2013, Fig.9). Please put into context.

(12) P.10,l.173-176: sure, nothing wrong with that. It remains, however, unclear, what the relevance of this is. We may be able to extrapolate the splitting coefficient for the future, but what exactly does the knowledge of this help us when future climatic forcing is unknown? Please clarify.

(13) P.11,l.198: should read as "...what was written..."

(14) P.11,eq.45: please clarify what the difference is to the relations discussed by Botter (2011) and Benettin et al. (2015)

[Figure]

(15) P.14,275ff, eqs.57-61: again, this is what is essentially done in most recent tracer based approaches. Please put into context.

(16) P.14,l.294-296: there are definitely obstacles to adequately determine SSF in reality. Not only due to uncertainties in precipitation and tracer input, but also due to oversimplified models and the related uncertainties (e.g. in Q) that typically lead to considerable equifinality (i.e. the well-known closure problem; Beven, 2006)

(17) P.14,l.304ff: sure, but not new. See for example the work of Bertuzzo et al. (2013) or Benettin et al. (2015).

(18) P.15,l.317: maybe mention that a linear reservoir here entails complete mixing/uniform SSF

(19) P.15,l.319: please clarify in detail what Rtau is.

(20) P.15,eq.63: as it is presented right now, this ignores the critical difference between celerity and velocity, or in other words that the response time distribution of a pressure wave routed through the system is significantly different to the travel time distribution of an actual input signal (McDonnell and Beven, 2014). This needs to be made clear!

(21) P.15, eq.68: this does essentially boil down to the convolution integral used in many earlier studies starting from the 1960s or so. Please put into context and highlight the relevance here.

(22) P.16,l.335: of course, as already argued by others previously (e.g. Rinaldo et al., 2011)

(23) P.18,l.423: should read as "...damped..."
* * *

---

## Referee Comment (RC2) · PB Benettin (Referee) · 14 Jun 2016

GENERAL COMMENTS

The paper offers an interesting perspective on the equations that govern water age evolution in catchments. Building upon previous works (in particular Botter et al., 2011, van der Velde et al., 2012, Harman 2015 and Benettin et al., 2015), the manuscript explores the hydrologic balance equation in the travel time dimension, and all the age distributions associated with it. Although the paper is written in formal mathematical language, some parts may be difficult to follow and should be expanded, possibly including some physical interpretation. I encourage the authors to improve the readability of the paper and I list below some suggestions.

DETAILED COMMENTS

**Notation**

Everyone is of course free to use the notation which is most suitable to explain their research. However, last year many researchers made a big effort to converge upon a unifying and useful notation for the travel time literature, based on their experience. The notation was presented in Harman, (2015) and Rinaldo et al., (2015), and has been used in many other papers. If the authors want to clarify some concepts in the travel time formulation, I think the use of a different notation doesn't help the reader. I can understand the use of a probabilistic approach and the conditional probability notation, but I don't see the need for changing symbol for the most elementary variables. In particular I refer to:

- $\tau$ instead of $t_{in}$ and $\iota$ instead of $t_{ex}$ , for the injection and exit time of a water parcel in the domain
- SSF instead of SAS, to denote the StorAge Selection functions
- The lack of a subscript $S$ to denote the age distributions that refer to the water storage

**Use of probability distributions**

The travel time literature was often related to stochastic hydrology and therefore it is natural to formulate its theory using probability formalism. However, in such a case, a probabilistic framework should be given i.e. one should define what the relevant random variables and the related sources of randomness are. Note, however, that the connection with stochasticity is often weak in applications. When one estimates the age distribution of streamwater at a certain time, he doesn't mean that there is a certain probability that water has a certain age. He wants to say that there is a distribution of water particles, each with a different estimated age, and those ages together can explain the measured solute concentration. Another example

pertains the marginal distributions: the "probability" of observing an input (or an output) is not taken from stochastic hydrology concepts, it is instead a normalized timeseries of actual precipitation (or discharge) measurements.

**Section 2**

Eq 1: I think it should be explained that the time variables refer to a parcel of water that moves inside a hydrologic volume, entering at a time $\tau$ and leaving at a time $\iota$.

Line 72: I have often seen this integral expressed between $-\infty$ and $t$. Maybe it would avoid the need to specify that time $t = 0$ comes before any input to the system?

Line 85: it may be worth saying that eq. (85) is a spatially-integrated equation that can be easily related to previous works in the literature, in particular Ginn (1999), Delhez et al., (1999) and Dagan (1984). See Benettin et al., (2013) for a review on this.

**Section 3** (backward and forward approaches)

I think this section is very important but not sufficiently developed. The authors present 4 different kind of probabilities and it may be worth to specify which probabilities pertain to the different elements of the water budget. In particular, when the time $t$ is to be interpreted as a general time at which the system is observed, the probabilities refer to the water particles in storage. Instead, when the time $t$ has the special meaning of time at which particles enter or leave the system, the probabilities refer to the water particles in the fluxes.

Line 90: Please if possible refer to the paper Benettin et al., 2015 published on Hydrological Processes, instead of the Ph.D. thesis Benettin 2015, as the latter has no DOI.

Line 91: this a bit imprecise: in the mentioned paper the concept of backward refers to the residence time (or age). Instead, the concept of travel time is both a forward or backward concept, depending on the point of view (i.e. if ones focuses on the entrance or on the exit).

Line 95-96: given your definition in eq. (1), $t - \tau$ is a residence time and not a travel time, so the distributions should be residence time probabilities and not travel time probabilities.

**Section 4**

This section, in my opinion, does need to be further explored. I recommend the authors to include a physical interpretation of the processes, besides the mathematical description. Also, I think it may help the reader if the probability distributions associated with the water storage were denoted with a subscript S (e.g. $p_S$), just like the authors did for the probability associated with discharge ($p_Q$) or evapotranspiration ($p_{E_T}$)

Line 105: Benettin (2015) used the notation $\breve{p}_S(T_R, t)$ and not $\breve{p}(T_R, t)$

Line 108-109: I did not understand this sentence (and maybe you meant eq (5) instead of eq (1))

Line 112: as this is a definition, the symbol $:=$ should be used?

Line 120: please enclose the first term at left hand side of eq. (14) within brackets, otherwise the derivative refers to $S(t)$ only. Please also explain what the symbol $\delta(t - \tau)$ refers to and why.

Line 129: please enclose the first term at left hand side of eq. (17) within brackets

Line 131: please specify that this is only valid in case eq. (17) is linear, i.e. $\omega(t, \tau)$ is not a function of $p(T_R|t)$

Line 135-145: Figures 2,3 and 4 need to be given more context and be better explained, because I had a hard time to interpret them and I am not sure the indicated $t$ and $\tau$ are all in place . Please also specify what $P(T_R|t)$ is. I think the main point is letting the reader understand what happens when one keeps the chronologic time $t$ fixed and let the injection time $\tau$ change, or vice versa.

**Section 5**

Just like Section 4, I think more interpretation should be provided. E.g. what does $p(t - \tau, \tau)$ represent? How is it related to $p_Q(t - \tau, \tau)$ and $p_{E_T}(t - \tau, \tau)$?

I think there may be some misunderstanding on the fact that the forward distributions cannot be known after time $t$. The problem is more general and equally applies to backward distributions, which are unknown for any time $t$ lower than the first available measurement. Moreover, unless one needs to do real-time predictions, travel time computations can be done on datasets which are much longer than the period of interest.

Line 1: Please explain what you mean by "integral form". It's important to specify that you are integrating over $dt$, hence you follow a single injection and track its evolution while crossing the catchment.

Line 68 (eq. 29): I don't see why the asymptotic value is that important. I believe the knowledge of $q(t, \tau)$ is much more important, and $\Theta(\tau)$ is just its integral over $dt$.

**Section 6**

Line 181: as the authors are focusing on a parcel of water that enters at time $t = \tau$ and exits at $t = \iota$, I think the equations should be rewritten using $\iota$ instead of $t$.

Line 187: the symbol $S$ for eq (34) and following is a bit ambiguous as the symbol $S(t)$ already appears at eq. (5) with a different meaning. Moreover, as the integral is defined from 0 to infinity, it appears to me that $S \rightarrow \infty$.

Please give an interpretation of the marginal pdf's.

I think the notation could be made more "symmetrical". Why is $p(\tau)$ the marginal pdf of the input and $p_Q(t)$ is the marginal probability of the output? Shouldn't they be either $p(\tau)$ and $p(\iota)$ or $p_Q(t)$ and $p_J(t)$?

Line 197: given the notation in eq. (37), it should be $p_Q(t)$ instead of $p(t)$

**Section 7**

In section 7 and 8 I found several typos and little errors in the formulas. I am not sure I detected them all, so please carefully revise these sections.

As in section 5, I think the authors should not assume by default that our knowledge limits to time $t$, and everything between $t$ and $\iota$ is unknown. That is just a very special case, limited to real time forward modeling.

Line 226: $\iota - t$ instead of $\iota - s$

Line 229: isn't the canonic convolution symbol different?

Line 235: typo: time time

Line 237: typo: $\tau$s

**Section 8**

Please specify that the balance equation is in terms of substance mass.

Line 251: please add the dependence on entrance time at left hand side of the equation

Line 261: I would suggest, for consistency with eq. (53) and (54) to name it $C_Q^i(t)$ (note in the table of symbols at the end of the manuscript it is listed $C_Q^i(t)$)

Line 263: I don't understand the expression "it is usually assumed the validity of an integral expression like…". Given the definitions of the terms involved in the equation, the integral is just an expression of mass conservation.

Line 265: $\Theta$ should be a function of $\tau$. Please note that here you use the product $\Theta(\tau)p(t - \tau|\tau)$ but in the subsequent equations $\Theta(\tau)$ disappears.

Line 271: shouldn't it be $p_Q(t - \tau|t)$ instead of $p(t - \tau|t)$? In your notation, the latter denotes the age probability of the water storage (eq. 10).

Line 287 and 294-296: why you did not use a dirac delta function (like in eq. 17) for the age distribution of precipitation?

Line 291-293: this sounds interesting but I am not sure it is actually feasible. I can't say what the issue may be from a mathematical point of view, but I have two examples in mind which are in conflict with the possibility of deriving $p(t - \tau|t)$ and $\omega(t,\tau)$ by coupling different budget equations. The first is that I am not sure the added mass budget equation is actually bringing different information with respect to the water budget equation. If the solute is conservative, it is transported just like water, so its concentration will simply be proportional to that of water, where $C_j^i(t) = 1$. The second is that in case of multi-solute information one could get a system with 3 or more equations and just two unknown functions, so following your reasoning it would be impossible to solve, and this sounds strange to me.

**Section 9**

I think there is a very important assumption here, which is not explicitly stated. The fact that discharge is a linear function of storage does not automatically mean that each parcel of discharge (from an age-rank point of view) is a linear function of the corresponding parcel in storage. This only happen when a well-mixed (o random sampling) scheme is assumed.

I think it is also important that the authors acknowledge here the difference between the storage which can be estimated from a purely hydrologic balance (i.e. the "active" storage which gets displaced during the

hydrologic response) and the total storage of a system. This is very important considering the recent debate about water displacement and water travel times (e.g. McDonnell and Beven, 2014, Rinaldo et al., 2015, Kirchner, 2016). If the equations derived in this section do not take into account the so called "passive" storage (Kirchner, 2009), they may be of limited practical use.

**Conclusions**

Line 394: typo "to the obtain understanding"

**Appendix A**

Please define $\Theta(t, \tau)$, as only $\Theta(\tau)$ has been defined so far, and use consistently the upper-case or lower-case symbols. Also, I guess some SSF for both discharge and evapotranspiration was imposed to produce the figures and I think it should mentioned.

REFERENCES

Benettin, P., Rinaldo, A., and Botter, G. (2015). Tracking residence times in hydrological systems: forward and backward formulations, Hydrological Processes.

Benettin, P., Rinaldo, A., and Botter, G. (2013). Kinematics of age mixing in advection-dispersion models, Water Resources Research.

McDonnell, J. J., & Beven, K. (2014). Debates - The future of hydrological sciences: A (common) path forward? A call to action aimed at understanding velocities, celerities and residence time distributions of the headwater hydrograph, Water Resources Research.

Kirchner, J. W. (2009). Catchments as simple dynamical systems: Catchment characterization, rainfall-runoff modeling, and doing hydrology backward, Water Resources Research.

Kirchner, J. W. (2016). Aggregation in environmental systems - Part 2: Catchment mean transit times and young water fractions under hydrologic nonstationarity, Hydrology and Earth System Sciences

Dagan, G. (1984). Solute transport in heterogeneous porous formations, Journal of Fluid Mechanics

Ginn, T. (1999). distribution of multicomponent mixtures over generalized exposure time in subsurface flow and reactive transport: Foundations, and formulations for groundwater age, Water Resources Research

Delhez, E. J. M., Campin, J.-M., Hirst, A. C., & Deleersnijder, E. (1999). Toward a general theory of the age in ocean modelling, Ocean Modelling

---

## Short Comment (SC1) · 5 Jul 2016

D. Wilusz

dwilusz1@jhu.edu

Received and published: 5 July 2016

Thank you for writing and sharing this manuscript. As a graduate student using storage selection functions and other transit time models, I benefited from seeing the theory synthesized and presented in a new way. Although I lack the depth of understanding of the reviewers and other experts in the field, I wanted to share some notes I made to myself as I read the paper, in case any of them can be helpful.

Line 7, line 33-34: I was intrigued by the idea of deriving SSF functions (1) for the Nash cascade (line 7) and (2) for relatively complex cases with real-world data (line 33-34), and would have enjoyed more development in both these areas of the manuscript.

Figure 2 - Should this be labeled as a 'residence time backward' cdf? What data was used to produce these and the other plots?
Figure 3 - What is the residence time backward pdf over a continuum of injection times (the x axis) for a single injection time (e.g., tau1)? Shouldn't tau1 just be a slice on the x-axis?

Line 155-160: Is there any problems with using equations 26-28 when rainfall is sometimes zero?

Line 187: Will this integrate to infinity? Or is the second "=" supposed to be "-"?

Line 193: The link to Bayes theorem is interesting. If Bayes theorem is  $(P(A|B)^*P(B) = P(A)^*P(B|A))$ , what are the equivalent events A and B in Niemi's relation?

Line 338-340: Aren't other (including time-varying) SSF functions possible even if discharge is proportional to storage?

---

## Author Comment (AC1) · 20 Jul 2016

**Comments of Reviewer #1**

We thank the reviewer for her/his observations, which helped us improve our manuscript. This response tries to use the practice of the interactive discussion, and does not yet produce a new manuscript, which will be submitted after the closure of the review phase, upon request of the Editor. We introduce here, however, the main adjustments that we will make in the final version of the revised paper based upon the reviewer's suggestions.

**R1 - The value of a travel time based description of catchment hydrology**

has been increasingly acknowledged over the recent years. Therefore, this manuscript comes timely and could eventually be an interesting contribution to literature. Presenting a considerable string of formalisms, describing various aspects of travel times, and which are, as far as I can see, mathematically sound, the authors delve deep into the topic.

A 1 - We thank the reviewer for this recognition.

R2 - However,even after reading the manuscript three times I struggle to see what the actual intended contribution is. What do the authors want to convey to the reader? This needs to be made much clearer. Is it a review of existing concepts? Is it an extension of existing concepts? If it is a review, the description of the concept needs to go further back toinclude earlier work and detailed descriptions thereof. If it is rather an extension of existing concepts, it needs to be clarified what the novelty is and how it fits into our current understanding. In other words, what are the main findings? What do we learn?

A2 - If the reviewer did not understand the main contribution of our work, it is certainly our fault, and we will be more clear in the revised version of the paper.

Our work is absolutely a short review of existing concepts that were collected from many (theoretical) papers where they were scattered and used not systematically. Besides presenting the concept in a new and organised way, our paper contains some, we believe non-trivial, clarifications and extensions.

The paper, as a proof of concept, includes one example derived from a real case (Posina river basin). Besides, our paper comes with open source code that implements the theory and is available to any researcher.

Clarifications include:

- The concepts of forward and backward probabilities (as conditional probabilities) and a small change in notation that should not be overlooked;

- their relation with the water budget (and the age-ranked functions) from which they were derived;

- the fact that time-invariant forward probabilities usually imply time-varying backward probabilities, i.e. travel time distributions.

- The rewriting of the BBR's (Botter, Bertuzzo and Rinaldo) master equation as an ordinary differential equation (instead of a partial differential equation).

- The role and nature of the partitioning coefficient between discharge and evapotranspiration (which is unknown at any time except asymptotically).

- The significance of the SSF (formerly called SAS) functions with examples.

- The relationship of the present theory with the well known theory of the instantaneous unit hydrograph.

Extensions include:

- New relations among the probabilities (including the relation between expectancy of life) and travel time probabilities.

- An analysis of the partitioning coefficients (which are shown to vary seasonally)

- An explicit formulation of the equations for solutes which would permit a direct determination of the SSF on the basis of experimental data.

- Test of the effect of various hypotheses (e.g. linear model of backward probability, and in the revised version, gamma model for the backward probabilities).

- In the revised version, we also add information and clarify some links of the present theory with Dehlez (1999) and Duffy (2010).

- Besides, answering to a question posed by the third reviewer has brought to an extension of Niemi's relation (and a new normalisation), which will be included in the revised version of the paper

- The presentation of Niemi's relation as a case of the Bayes Theorem.

- A system of equations from which to obtain the SSFs.

**R3 - In either case, the presented work needs to be put into a broader context. The authors refer to a few key publications, but they ignore many other recent contributions that address the issue from similar and/or different perspectives. Such a broader context will help the reader to better appreciate the relevance of the presented work. I would thus invite the authors to discuss their methods and findings with respect to methods, results and findings from a wider range of other (also more experimental) studies including, for example Birkel et al. (2010, 2011, 2014), Fenicia et al. (2010), Van der Velde et al. (2010, 2015), McMillan et al. (2012), Hrachowitz et al. (2015), Rinaldo et al. (2015) but also with the work of Cvetkovic, Fiori, Dagan, etc over the past years.**

A3 - We gladly accept the suggestions of the reviewer, and will provide a more extensive treatment of the subject in the revised paper, which will include the papers she/he cites and a few others. As a general comment, we remind the reviewer that the large part of the literature on the subject we read is based on the (very limiting)

hypothesis of stationary velocity fields, and therefore travel time distributions (TTD) are time-invariant, while our theory is centered on a time-variant approach, which we claim is unavoidable. This can be derived from the water budget coupled with BBR's master equation.

To our knowledge, there are no clear contributions before 2012 in our discipline's literature, and the findings we collect are scattered among various papers. Delhez's group at Lovain (e.g., Delhez, 1999, and Delhez et al., 2001) actually proposed a formalism that is similar to ours but it used concentration instead of probabilities, added creation destruction terms, and uses a completely different parameterisation of the fluxes, in such a way that is hard to recognise that its equation (4) is, in concept, our equation (9). Delhez's theory was more recently used in our contexts by Duffy (2010) who made clear that actually the equation can form dynamical systems whose solution estimates conjointly concentrations and ages. His formalism is foreseen to be compatible with ours, and we will add some phrase about it in the revised manuscript. Our equation is also equivalent to the one presented in Ginn (1999), e.g. equations (10) and (12), but again in a form which is far from our notation and concepts, and not easily understood. Carrera (1998) remarkable work, can be commented the same way. Going back into the literature, Campana (1987) also wrote an equation for water age distribution. He used a discrete time formalism, that is also not easily translatable into our derivation. We will add part of this information in the revised version of the paper.

**Other comments:**

**(1) P.1, l.17-18: there is, in my understanding, little that remains unclear. Perhaps provide some examples. (2) P.2, l.26: again, I sort of disagree, there is little that remains unexplained. Please also give an example here to clarify.**

Recent contributions on time-variant distributions in groundwater hydrology include Alì

et al., 2014, Cvetkovic et al., 2014, and Soltani and Cvetkovic, 2013. Notably Soltani and Cvetkovic (2013) made a small mistake, in which their Figure 3 has the actual time in abscissa, while it cannot be. In fact, as it results clearly from our formalism, backward probabilities are functions of the actual time, and forward probabilities are functions of the injection time. Both can be seen, instead, as functions of the "travel time". It could be just a negligible oversight, but it is possibly a sign that the formal part of the theory and the related concepts were not well assessed before our paper. In fact, in other papers that inspired us, there are other many such small imprecisions that obfuscate the understanding of the reader, such as: confusing joint probability with conditional probability; taking for granted the knowledge of the partition coefficient when multiple outlet are present; and integrals that should be limited to the actual time t, go to infinity instead. Figure 4 of Hrachowitz, 2013, where one of the formulas is incorrect, is another example.

We will try to convey all this information better in the revised version of the manuscript. In general, we remark that all was already done is actually not true. Moreover, we think that discovering the concepts we clarify in the original paper could have been daunting even for a trained reader as the reviewer certainly is.

**(3) P.2, l.33-41: please be more specific here: what is the research hypothesis to be tested?**

In lieu of a general hypothesis statement, we pose the following questions: does the theory of Travel Times, as developed in recent years constitute a consistent unique framework ? Does it have hidden parts that there are not consistent or unexplained ? How it relates to the instantaneous unit hydrograph theories ? How can it be used ? What generates time varying backward probabilities ?

The answers we present are:

- The theory is consistent and gives a complete and unique view of the travel time theory.

- There are parts that are not fully explained (the role of the partitioning coefficient, for instance).

- The theory can be used once the water budget is solved. However, in this case, probabilities remain unknown until time goes toward infinity.

- On the contrary, as traditionally pursued in the instantaneous unit hydrograph theory, when forward probability is given, all is known; but with probability choice we make a precise statement about the future.

- Time varying backward probability are easily generated by time varying rainfall (and are therefore unavoidable), even in the classical case of systems described by time-invariant forward probabilities.

**(4) P.2,l.47 and 48: this should read as ". . .the time at which. . ." to avoid confusion, as we are (as a simplifying assumption) not talking about a time interval over which the input occurs but an instantaneous input.**

We corrected the paper accordingly to the suggestion of the reviewer.

**(5) P.3,l.57-58: please add the respective dimensions**

Done

**(6).1 P.3,l.67: this can only be solved analytically if piecewise linear functions of inputs are available and, more importantly, with the assumption of only one storage component in the system, which may be quite an oversimplification for most catchments.**

We agree with the reviewer.

**The water balance as given here cannot resolve the non-linearites in the system, including interception, vadose zone dynamics, storm flow connectivity, etc.. Thus the practical utility of such an analytical solution, if not used in an operator splitting strategy that accounts for different system components, remains limited. Please qualify the statement accordingly.**

The generalisation to multiple cascading storages is quite obvious, but more complicated to explain. This is the reason we limited ourselves to a simple system. For more complicated and interconnected systems the water budget must be solved numerically. It is actually written at line 68, but we will made it more clear and general. When more than one reservoir is present, we have a set of budget equations to be solved simultaneously. We will write more clearly these concepts in the revised manuscript.

**(7) P.4,l.86: Really? I would be surprised by that as this is implicit (and has to be) in essentially all approaches that somehow track fluxes through the system. I could imagine that this has already been explicitly formulated earlier. Please check, in particular papers by Cvetkovic, Dagan, Fiori, Russo, etc.!**

Yes. We already partially answered to this question in the above points. We went back to the literature in various fields. Even if we share the reviewer incredulity about the result, the answer is, to our knowledge, that there is no trace of equation (9) in

papers of the authors he/she mentions (or others, to our knowledge). Certainly many authors treated advection-dispersion equations: but only very recently they moved to treat time-dependent travel time probabilities. The original formalism (Dagan, 1989) is really general, so it contains what equation (9) conveys. But it does it as Navier-Stokes equation is contained in Newton's second law of dynamics: a lot of assumptions and mathematical treatment have to be done to pass from the latter to the former. Obviously, there is always the possibility that we overlooked some contribution.

**(8) P.5, section 4: that is all fine and true, but nothing new. It remains unclear what the purpose of this section is. Please clarify!**

Without the definitions contained in this section the paper loses its validity. To remark what the backward probability is and how it relates to the quantity of the water budget is deemed essential to understand the whole structure of the theory, even if the result that differentiates our contribution from the others is that the BBR's master equation is an ordinary differential equation in our derivation, instead of a partial differential equation. This was obtained by considering the explicit form of the travel time variable and exposing the dependence of the equation on the injection time. One original result is also represented in equation (21), which is the fact that the backward probability, thought as a function of t (domain in which it is not a probability) is null when it is not raining. In itself this is a little theorem, that we could have added as an extension in the list of answer A2.

**(9) P.6,l.136: I may have missed something here, but how can P(Tr/t) not integrate to one (and it seems it actually does in figure 2)?? As far as I understand, it is the sum of storages of all given ages present in the system over the total storage at any time t. Also: how does ageing contribute here? Please clarify.**

The integral of $p(t-\tau|t)$ is equal to 1 only if integrated in $\tau$. When it is integrated in t, its integral is less than 1. The notation used in many papers (without the symbol | to indicate "conditional to") does not help to understand this. However, Figure 2 may not be clear, since in the abscissa is named "Time" and it should be "Injection time" instead. We will change it in the revised version of the manuscript.

**(10) Figures 2 and 4: I am a bit confused by this figure. How can three injections at three different injection times (tau1, tau2, tau3) plot on top of each other when the x-axis is the actual time t measured by a clock? Should here the x-axis not rather be the time since injection?**

This is true, their origin was shifted to coincide for comparison. Otherwise they would have a different origin. We will modify the caption of the Figure to make it clear.

**(11) P.8,l.161ff: that is correct, but has been shown and discussed earlier (e.g. Benettin et al., 2015, Fig.6; Hrachowitz et al., 2013, Fig.9). Please put into context.**

The concepts expressed at line 161 ff and in the two examples cited by the reviewer are completely unrelated. Fig 6 in Benettin 2015 shows that their estimates of the hydrological and transport parameters show a consistency across different simulations. In particular they could compare the parameters' posterior distributions resulting from the calibration of individual years, as an independent verification of the reliability of the calibration algorithm. Fig 9 in Hrachowitz, 2013 instead shows the results of the transit time distributions for the outflows considered in 4 different hydrologic regimes and for the 3 case studies. What we are saying instead is simply that at an finite time we do not know the shape of the forward distribution. What we know is only the actual state of the system, obtained solving the budget up to the actual time.

**(12) P.10,l.173-176: sure, nothing wrong with that. It remains, however, unclear, what the relevance of this is. We may be able to extrapolate the splitting coefficient for the future, but what exactly does the knowledge of this help us when future climatic forcing is unknown? Please clarify.**

The importance of $\Theta$ is well described in the Appendix A that we decided to move back into the main text in the revised version of the paper. The knowledge of the partitioning coefficient is important to characterise the basin response.

Besides it is important in the Niemi's relation between the backward and forward pdfs. As shown in Appendix A, $\Theta$ tends to a final value after a time, characteristic of the basin. Besides $\Theta$ is shown to vary seasonally, which is kind of obvious. This fact adds time variability to catchments responses which is usually neglected or accounted for with less clean methods.

**(13) P.11,l.198: should read as ". . .what was written. . ."**

Done.

**(14) P.11,eq.45: please clarify what the difference is to the relations discussed by Botter (2011) and Benettin et al. (2015)**

No difference. We added a citation to them.

**(15) P.14,275ff, eqs.57-61: again, this is what is essentially done in most recent tracer based approaches. Please put into context.**

This is true only for equations (57) and (58) (in fact we cited Rinaldo 2011). The next

equations are conceptually simple but they were derived from our new formalism. The derivation is so straightforward that we would be surprised if it is new. However, as matter of fact, they were not used in recent papers in the literature. They can be used to determine or at least to constrain the form the SSF functions. Parallel to Duffy (2010), we can show that we could avoid to use the SSF by simply inferring $p_Q(T_r|t)$ directly, by evaluating simultaneously the water and the tracer budgets. In any case, we believe that using the SSF adds knowledge of the hydrologic mechanisms. Some phrase about the direct determination of the $p_Q(T_r|t)$ will be added in the final version of the manuscript.

**(16) P.14,l.294-296: there are definitely obstacles to adequately determine SSF in reality. Not only due to uncertainties in precipitation and tracer input, but also due to oversimplified models and the related uncertainties (e.g. in Q) that typically lead to considerable equifinality (i.e. the well-known closure problem; Beven, 2006).**

Any tracer measurement (we would say any hydrologic measurement) is difficult to obtain, but this does not prevents experimental hydrologists from taking measurements and all of us try to infer knowledge from them.

Equation (61) is the tracer budget, and evaluating tracer inputs and outputs is what people in the field do daily. We cannot so easily say that their work is vain or useless. The SSF function (or the backward probability of discharges, as we remark in answer to comment 15) is just obtained by algebraic manipulation. The case we exploit (with just one storage) is purely illustrative, but generalisation to more complex systems, as those presented in Hrachowitz et al., 2013, is straightforward. In the revised manuscript we will add an explaination of how to do it.

**(17) P.14,l.304ff: sure, but not new. See for example the work of Bertuzzo et al.**

**(2013) or Benettin et al. (2015).**

We disagree. What we are saying here is that coupling the two equations (water and solute transport) we are able to determine the SSF exactly. In the cited studies, SSF values were imposed a-priori and the results checked a-posteriori. In particular Benettin et al. (2015) states: "*Despite the intuitive essence of the formulation and the advances achieved by using a transformed travel time domain [van der Velde et al., 2012; Harman, 2015], the use of SAS functions is still a challenge, because few real-world applications have been proposed in the literature and numerical solutions can be computationally demanding. A compelling alternative is to model the catchment through a series of physically meaningful storage partitions (typically, one for the shallow soil and one for deeper groundwaters) and assume a random sampling (RS) mixing scheme within each storage. This enables the use of analytical solutions that are particularly easy to implement. Moreover, the RS assumption was shown to give reasonable results in systems with high degrees of heterogeneity [Benettin et al., 2013a; Ali et al., 2014] and has been successfully applied to different settings [Bertuzzo et al., 2013; Benettin et al., 2013b] including comparisons to spatially distributed 3-D numerical models [Rinaldo et al., 2011]. Under the RS approximation, the SAS function is equal to unity, hence the travel and residence time distributions coincide [Botter, 2012; Hrachowitz et al., 2013]*" etc. Clearly Benettin's very recent paper is looking for a feasible method for obtaining SSF. We gave one more.

**(18) P.15,l.317: maybe mention that a linear reservoir here entails complete mixing/uniform SSF**

The fact that a linear reservoir implies complete mixing is written after line 338. In the same section it is also shown that if the set of linear reservoirs have a mean travel time $\lambda_\tau$ which is dependent on the injection time, there is not complete mixing.

**(19) P.15,l.319: please clarify in detail what $R_\tau$ is.**

It is already clarified below equation (63). Moreover it is also explained in the List of symbols.

**(20) P.15,eq.63: as it is presented right now, this ignores the critical difference between celerity and velocity, or in other words that the response time distribution of a pressure wave routed through the system is significantly different to the travel time distribution of an actual input signal (McDonnell and Beven, 2014). This needs to be made clear!**

The example used in this section is just a classical, commonly known, example. The first author recently argued about the concepts of wave celerity and water speed (e.g. Rigon R., Celerity versus velocity and the travel time problem,http://abouthydrology.blogspot.it/2016/06/celerity-vs-velocity.html, last retrieved 2016-07-04) and concluded that all the effects of pressure waves (which travel with a celerity different from the velocity of water) are the (only) cause of time-varying travel time backward distributions. However, in the example presented here, what is time-invariant is not the backward probability, but the forward one. The rationale of using it in the paper is to show how its choice determines the SSF (to be 1) and the mean travel time. Subsequently the hypothesis of time invariant linear storages is relaxed, by using time variant linear storages to better reveal the nature of the SSF functions. It is also shown that, a time invariant forward probability does not imply a time invariant backward probability, which is obtained below at equation (69). This fact is quite unexpected and, accordingly to the heuristic in Rigon (2016) cited above, implies the existence of a travelling signal that alters the stationarity of the velocity field. This will be made clearer in the final version of the manuscript.

**(21) P.15, eq.68: this does essentially boil down to the convolution integral used in many earlier studies starting from the 1960s or so. Please put into context and highlight the relevance here.**

That's true, equation (68) is the convolution of effective rainfall impulses with the travel time distribution. What is not trivial here is to note that the backward probability (the one that affects tracers) is actually dependent on the amount and timing of rainfall input. Therefore, even in this case, where we have a time invariant forward probability, we obtain a time-variant backward probability. This time variation is indeed trivial because it has an obvious dependence on the rainfall inputs. However, both the numerator and denominator of the equation depend on the rainfall inputs, and so the time-variation scheme of the backward probability can be understood but not simplified (for instance by factorizing the rainfall input). We will put all of this in context and we will highlight it better in the revised version of the manuscript.

**(22) P.16,l.335: of course, as already argued by others previously (e.g. Rinaldo et al., 2011)**

After more thought, yes, it is a result that is implied in Rinaldo et al., 2011. We will add a citation here.

**(23) P.18,l.423: should read as ". . .damped. . ."**

Corrected accordingly

**References**

Ali, M., A. Fiori, D. Russo, A comparison of travel-time based catchment transport models, with application to numerical experiments, Journal of Hydrology, 511, pg. 605-618, http://dx.doi.org/10.1016/j.jhydrol.2014.02.010, 2014.

Benettin, P., J. Kirchner, A. Rinaldo, and G. Botter (2015), Modeling chloride transport using travel-time distributions at Plynlimon, Wales, Water Resour. Res., 51, 3259–3276, doi:10.1002/2014WR016600.

Birkel, C., D. Tetzlaff, S. M. Dunn, and C. Soulsby (2010), Towards simple dynamic process conceptualization in rainfall-runoff models using multi-criteria calibration and tracers in temperate, upland catchments, Hydrol. Processes, 24, 260–275.

Birkel, C., D. Tetzlaff, S. M. Dunn, and C. Soulsby (2011a), Using time domain and geographic source tracers to conceptualize streamflow generation processes in lumped rainfall-runoff models, Water Resour. Res., 47, W02515, doi:10.1029/2010WR009547.

Birkel, C., C. Soulsby, and D. Tetzlaff (2011b), Modelling catchment scale water storage dynamics: Reconciling dynamic storage with tracer inferred passive storage, Hydrol. Processes, 25, 3924–3936.

Birkel, C., C. Soulsby, and D. Tetzlaff (2014), Developing a consistent process-based conceptualization of catchment functioning using measurements of internal state variables, Water Resour. Res., 50, 3481–3501, doi:10.1002/2013WR014925.

Campana, M. (1987). Generation of ground-water ge distributions. Groundwater, 25(1), 51–58.

Carrera, J. (1998). Simulation of groundwater age distributions. Water Resources Res., 34(12), 3271–3281.

Cvetkovic, V., Carstens, C., Selroos, J.O., Destouni, G., 2012. Water and solute transport along hydrological pathways. Water Resour. Res. 48, W06537. http://dx.doi.org/10.1029/2011WR011367.

Delhez, E.J.M., Campin, J.M., Hirst, A.C., Deleersnijder, E., 1999. Toward a general theory of the age in ocean modelling. Ocean Mod. 1, 17–27.

Duffy, C.J., 2010. Dynamical modelling of concentration–age–discharge in watersheds. Hydrol. Process. 24 (12), 1711–1718.

Fenicia, F., Wrede, S., Kavetski, D., Pfister, L., Hoffmann, L., Savenije, H. H. G., & McDonnell, J. J. (2010). Assessing the impact of mixing assumptions on the estimation of streamwater mean residence time. Hydrological Processes, 24(12), 1730–1741. http://doi.org/10.1002/hyp.7595

Ginn, T. (1999). Distribution of multicomponent mixtures over generalized exposure time in subsurface flow and reactive transport: Foundations, and formulations for groundwater age, Water Resources Research

Hrachowitz, M., Savenije, H., Bogaard, T. A., Tetzlaff, D., & Soulsby, C. (2013). What can flux tracking teach us about water age distribution patterns and their temporal dynamics? Hydrology and Earth System Sciences, 17(2), 533–564. http://doi.org/10.5194/hess-17-533-2013

McMillan, H. K., D. Tetzlaff, M. Clark, and C. Soulsby (2012), Do time-variable tracers aid the evaluation of hydrological model structure? A multimodel approach, Water Resour. Res., 48, W05501, doi:10.1029/2011WR011688.

Nauman, E.B., 1969. Residence time distribution theory for unsteady stirred tank reactors. Chem. Eng. Sci. 24 (9), 1461.

Soltani, S.S., Cvetkovic, V., 2013. On the distribution of water age along hydrological pathways with transient flow. Water Resour. Res. 49. http://dx.doi.org/10.1002/wrcr.20402.

van der Velde, Y., G. H. de Rooij, J. C. Rozemeijer, F. C. van Geer, and H. P. Broers (2010), Nitrate response of a lowland catchment: On the relation between stream concentration and travel time distribution dynamics, Water Resour. Res., 46, W11534, doi:10.1029/ 2010WR009105.

Zuber, A., 1986. On the interpretation of tracer data in variable flow systems. J.Hydrol. 86 (1–2), 45–57.

---

## Author Comment (AC2) · 20 Jul 2016

**Answer to reviewer #3, Mr. D. Wilusz**

**Thank you for writing and sharing this manuscript. As a graduate student using storage selection functions and other transit time models, I benefited from seeing the theory synthesized and presented in a new way. Although I lack the depth of understanding of the reviewers and other experts in the field, I wanted to share some notes I made to myself as I read the paper, in case any of them can be helpful.**

We thank Mr. D. Wilusz for his reading. We appreciate his observations and comments, and we believe that they will result in effective improvements of the final manuscript.

[Figure]

**Line 7, line 33-34: I was intrigued by the idea of deriving SSF functions (1) for the Nash cascade (line 7) and (2) for relatively complex cases with real-world data (line 33-34), and would have enjoyed more development in both these areas of the manuscript.**

We will add more information to allow everybody to understand complex storage combinations. By the way, our source code, which uses more than one storage for producing the probability figures of the paper, is open source. It is available at https://github.com/geoframecomponents and commented on http://geoframe.blogspot.com

**Figure 2 - Should this be labeled as a 'residence time backward' cdf? What data was used to produce these and the other plots?**

We changed the notation according to the review of Dr. Benettin (reviewer #2), introducing the subscript $S$ to denote the storage and thus the residence time distributions. Now the figure makes it clearer that we are talking about the backward residence time cdf. The figures were obtained using Posina River data. In particular, the data available were rainfall, temperature and discharge time series. Thanks to the model JGrass-NewAge we were able to delineate the HRUs, handle spatial data, simulate the radiation balance (shortwave and longwave), estimate the ETp and simulate the discharge. The solution of the water budget is also implemented.

**Figure 3 - What is the residence time backward pdf over a continuum of injection times (the x axis) for a single injection time (e.g., tau1)? Shouldn't tau1 just be a slice on the x-axis?**

The residence time is a backward pdf, which properly integrates to a value of one (1) when it is integrated in the injection time dimension. Both figures 2 and 4 show the

evolution of the previous pdfs. In the first case (figure 2) we kept fixed the chronological time and let the injection time vary; in this case, the integral of the area under the curves (figure 3) is 1. Vice-versa, in figure 4 we kept fixed the injection time and let the chronological time to vary. We are going to explain the figures more thoroughly in the new version of the manuscript, according to the reviews.

**Line 155-160: Is there any problems with using equations 26-28 when rainfall is some- times zero?**

As we show in equation (21) and in figure 4, the probability remains constant when rainfall is not present and t varies.

**Line 187: Will this integrate to infinity? Or is the second "=" supposed to be "-"? Line 193: The link to Bayes theorem is interesting. If Bayes theorem is $(P(A|B) * P(B) = P(A) * P(B|A))$, what are the equivalent events A and B in Niemi's relation?**

We answer the two questions together. The equation is properly written. It integrates to infinity, and the second symbol is an "=". We will try to clarify this point better. S represents the net precipitation (input), which is equal to the total discharge (output). So one probability is the probability density of a single precipitation event, and the other is the probability density of a single discharge instant. However, we realised that this definition has a drawback that needs to be solved and brings to generalise Niemi's relation. The problem with the above definition is that any density is zero when the time domain tends to $\infty$. However, the forward probability can be re-defined in a more general way to restrict our consideration to precipitations fallen in a finite interval $[0, t']$. For any $t \leq t'$ the algebra and the definition of the quantities in the Niemi's relation follows what already presented. Instead for $t > t'$, i.e. for actual time interval when rain

is not falling or is being neglected, normalisation changes and:

$$p(t - \tau | t, \tau \in [0, t']) := \frac{q(t, \tau)}{Q_{t'}(t)} \quad \tau \in [0, t'] \tag{1}$$

and:

$$Q_{t'}(t) := \int_0^{t'} q(t, \tau) d\tau \tag{2}$$

Therefore $Q_{t'}(t)$ represents the fraction of discharge at time $t$ generated by precipitation fallen for all the $\tau \in [0, t']$. At this point a generalised Niemi's relation holds as:

$$Q_{t'}(t) p(t - \tau | t, \tau \in [0, t']) = \Theta(\tau) p_q(t - \tau | \tau) J(\tau) \quad \tau \in [0, t'] \tag{3}$$

After restricting to a limited set of precipitations, the total amount of precipitation is:

$$S_{t'} = \int_0^{t'} J(\tau) d\tau \tag{4}$$

from which we have:

$$\frac{Q_{t'}(t)}{\Theta(\tau) S_{t'}} p(t - \tau | t, \tau \in [0, t']) = p_q(t - \tau | \tau) \frac{J(\tau)}{S_{t'}} \tag{5}$$

Defining:

$$p(\tau | \tau \in [0, t']) := \frac{J(\tau)}{S_{t'}} \tag{6}$$

and

$$p_Q(t | \tau \in [0, t']) := \frac{Q_{t'}(\tau)}{\Theta(\tau) S_{t'}} \tag{7}$$

We obtain again the Bayes theorem where $p(\tau | \tau \in [0, t'])$ is the fraction of the precipitation fallen at time $\tau$ with respect to the whole precipitation fallen $\tau \in [0, t']$. Similarly,

$p_Q(t|\tau \in [0, t'])$ is the fraction of discharge at time t generated by precipitation fallen at $\tau$. No precipitations fallen before or after the interval $[0, t']$ count, and the relative discharge has to be emended in the calculations. Remarkably, even if we would not be interested in the Bayes theorem analogy, the extended formalism properly identifies which are the normalisation factors to be used in defining the backward probabilities.

In the revised version of the paper we will modify the section about the Niemi's relation to account for the above considerations.

**Line 338-340: Aren't other (including time-varying) SSF functions possible even if discharge is proportional to storage?**

No, it follows from the definition of the backward probabilities that:

$$p(t - \tau|t) := \frac{s(t,\tau)}{S(t)} = \frac{\lambda q(t,\tau)}{\lambda Q(t)} = \frac{q(t,\tau)}{Q(t)} =: p_Q(t - \tau|t) \tag{8}$$

This happens because $\lambda$ is not dependent on $\tau$. Viceversa, as happens in equation (71) of the paper, the equality above does not hold because each injection time has its own ($\tau$ dependent) $\lambda$ (which is, in fact, labeled $\lambda_\tau$).

---

## Author Comment (AC3) · 20 Jul 2016

**Answer to reviewer #2, Dr. P. Benettin**

We thank Dr. Paolo Benettin for his detailed comments. This will help us to improve the manuscript.

**GENERAL COMMENTS**

The paper offers an interesting perspective on the equations that govern water age evolution in catchments. Building upon previous works (in particular Botter

et al., 2011, van der Velde et al., 2012, Harman 2015 and Benettin et al., 2015), the manuscript explores the hydrologic balance equation in the travel time dimension, and all the age distributions associated with it. Although the paper is written in formal mathematical language, some parts may be difficult to follow and should be expanded, possibly including some physical interpretation. I encourage the authors to improve the readability of the paper and I list below some suggestions.

We will try to add further explanations, trying be clearer and to improve the readability of the paper in the revised version.

**DETAILED COMMENTS**

**Notation**

Everyone is of course free to use the notation which is most suitable to explain their research. However, last year many researchers made a big effort to converge upon a unifying and useful notation for the travel time literature, based on their experience. The notation was presented in Harman, (2015) and Rinaldo et al., (2015), and has been used in many other papers. If the authors want to clarify some concepts in the travel time formulation, I think the use of a different notation doesn't help the reader. I can understand the use of a probabilistic approach and the conditional probability notation, but I don't see the need for changing symbol for the most elementary variables. In particular I refer to:

- instead of tin and instead of tex for the injection and exit time of a water parcel in the domain

- SSF instead of SAS, to denote the StorAge Selection functions

- **The lack of a subscript S to denote the age distributions that refer to the water storage**

We corrected the notation according to reviewer's request.

**Use of probability distributions**

**The travel time literature was often related to stochastic hydrology and therefore it is natural to formulate its theory using probability formalism. However, in such a case, a probabilistic framework should be given i.e. one should define what the relevant random variables and the related sources of randomness are. Note, however, that the connection with stochasticity is often weak in applications. When one estimates the age distribution of streamwater at a certain time, he doesn't mean that there is a certain probability that water has a certain age. He wants to say that there is a distribution of water particles, each with a different estimated age, and those ages together can explain the measured solute concentration. Another example pertains the marginal distributions: the "probability" of observing an input (or an output) is not taken from stochastic hydrology concepts, it is instead a normalized timeseries of actual precipitation (or discharge) measurements.**

Here, Dr. Benettin supports the idea that some variables (travel time, residence time, and life expectancy) can have a distribution and, in reality, the physics are not random, but deterministic, and the distributions are simply due to some causal dynamics. There are a few difficulties to accept his point of view. Reality is reality, and mathematics (the model) is mathematics (the model). All the probability distribution functions we use obey all the axioms required to be called probability. What the theory we presented

does is to try to constrain the limits where the choice is random. If we assumed well-mixed waters (SSF $\omega(t, \tau) = 1$), there is no doubt in the literature that we pick particles of water at random, without thinking of their age. If (SSF $\omega(t, \tau) \neq 1$) we reduce our (uniform) random sampling to the set of water of the same age, while we select the proportion of each age over the total storage. We agree with Dr. Benettin that hydrology could be ultimately different from the model. But there is no doubt that the model (we use and he, too, in his papers) obeys the axioms of probability and is based on random sampling.

**Section 2**

**Eq 1: I think it should be explained that the time variables refer to a parcel of water that moves inside a hydrologic volume, entering at a time and leaving at a time .**

Changed.

**Line 72: I have often seen this integral expressed between $-\infty$ and t. Maybe it would avoid the need to specify that time t=0 comes before any input to the system?**

We put the 0 limit to be more close to applications where the modeler has to chose an instant of time at which to start the modelling. After the comments of reviewer #3 we were actually push to change the notation to well defined time intervals.

**Line 85: it may be worth saying that eq. (9) is a spatially-integrated equation that can be easily related to previous works in the literature, in particular Ginn (1999),**

**Delhez et al., (1999) and Dagan (1984). See Benettin et al., (2013) for a review on this.**

We will add information about in the revised manuscript. However, the above statement is true for Ginn and Dehlez, but not for Dagan (an extraordinarily beautiful paper). As we wrote in the answer to reviewer #1, Ginn equation (10) and Dehlez equation (4) convey the same concepts with a very different formalism, and it is not so easy to grasp that actually they are talking about the same equation. Dagan's paper, instead, talks about stationary fields. So our paper, dealing with non-stationary situations, is the integrated form of a generalisation of Dagan's result.

**Section 3 (backward and forward approaches)**

**I think this section is very important but not sufficiently developed. The authors present 4 different kind of probabilities and it may be worth to specify which probabilities pertain to the different elements of the water budget. In particular, when the time t is to be interpreted as a general time at which the system is observed, the probabilities refer to the water particles in storage. Instead, when the time t has the special meaning of time at which particles enter or leave the system, the probabilities refer to the water particles in the fluxes.**

We added further explanations after the definition of the travel time and evapotranspiration time pdfs.

**Line 90: Please if possible refer to the paper Benettin et al., 2015 published on Hydrological Processes, instead of the Ph.D. thesis Benettin 2015, as the latter has no DOI.**

Done

**Line 91: this a bit imprecise: in the mentioned paper the concept of backward refers to the residence time (or age). Instead, the concept of travel time is both a forward or backward concept, depending on the point of view (i.e. if ones focuses on the entrance or on the exit).**

We corrected the sentence and added further explanations.

**Line 95-96: given your definition in eq. (1), $t - \tau$ is a residence time and not a travel time, so the distributions should be residence time probabilities and not travel time probabilities.**

Right. We corrected the mistake.

**Section 4**

**This section, in my opinion, does need to be further explored. I recommend the authors to include a physical interpretation of the processes, besides the mathematical description. Also, I think it may help the reader if the probability distributions associated with the water storage were denoted with a subscript S (e.g. $p_S$), just like the authors did for the probability associated with discharge ($p_Q$) or evapotranspiration ($p_{ET}$).**

We changed the notation accordingly.

**Line 105: Benettin (2015) used the notation $\overleftarrow{p_S}(T_r, t)$ and not $\overleftarrow{p}(T_r, t)$**

Corrected accordingly.

**Line 108-109: I did not understand this sentence (and maybe you meant eq (5) instead of eq (1))**

Corrected accordingly.

**Line 112: as this is a definition, the symbol := should be used?**

Yes, corrected accordingly.

**Line 120: please enclose the first term at left hand side of eq. (14) within brackets, otherwise the derivative refers to S(t) only. Please also explain what the symbol $\delta(t - \tau)$ refers to and why.**

Corrected accordingly.

**Line 129: please enclose the first term at left hand side of eq. (17) within brackets**

Corrected accordingly.

**Line 131: please specify that this is only valid in case eq. (17) is linear, i.e. $\omega(t, \tau)$ is not a function of $p(T_r|t)$**

Corrected accordingly.

[Figure]

**Line 135-145: Figures 2,3 and 4 need to be given more context and be better explained, because I had a hard time to interpret them and I am not sure the indicated t and are all in place . Please also specify what $P(T_r|t)$ is. I think the main point is letting the reader understand what happens when one keeps the chronological time t fixed and let the injection time change, or vice versa**

The figures were changed accordingly to both the #1 and #2 reviewer's observation, further explanation added and all the variables updated according to the new notation.

**Section 5**

**Just like Section 4, I think more interpretation should be provided. E.g. what does $p(t - \tau, \tau)$ represent? How is it related to $p_Q(t - \tau, \tau)$ and $pET(t - \tau, \tau)$?**

Maybe we are pedantic about this, but the notation is $p(t - \tau|\tau)$ which different from $p(t - \tau, \tau)$. The first expression in standard probability notation means a conditional probability, the second a joint probability. They convey different concepts and return different numbers, so they should not be confused. In this case the first is a forward probability, since it is conditioned to the injection time. In particular, as it should be clear from the notation, the first refers to the forward distribution of the residence time, while the latter expression once rewritten properly as $p_Q(t - \tau|\tau)$ and $p_{ET}(t - \tau|\tau)$, refers to the forward distribution of travel and evapotranspiration time. The three are related through the partitioning coefficient, which is another one of the main aspects in the forward approach.

**I think there may be some misunderstanding on the fact that the forward distributions cannot be known after time t. The problem is more general and equally**

**applies to backward distributions, which are unknown for any time t lower than the first available measurement. Moreover, unless one needs to do real-time predictions, travel time computations can be done on datasets which are much longer than the period of interest.**

We don't see the misunderstanding here. Instead the reference to real-time simulations and ex-post simulations is a little misleading. The main problem, that maybe we did not convey properly, is about causation. If the age-ranked approach is deployed on the basis of the solution of the water budget equation, from the definitions we gave it follows that the backward probability is completely known up to actual time t (and, obviously, if no measurement or estimation is made, nothing can be known). Whatever happens after the actual time, is not known (and should not, unless all the future story of discharges and evapotranspiration would). The same happens for the forward probability. The latter, however, has a further element unknown, which is the partition coefficient, which is known only at infinity. We will try to make this point more clear in the revised manuscript.

**Line 1: Please explain what you mean by "integral form". It's important to specify that you are integrating over dt, hence you follow a single injection and track its evolution while crossing the catchment.**

We added an explanation about the integral over dt.

**Line 168 (eq. 29): I don't see why the asymptotic value is that important. I believe the knowledge of $q(t, \tau)$ is much more important, and $\Theta(\tau)$ is just its integral over dt.**

The importance of the partitioning coefficient is explained in Appendix A. Its asymptotic values, as shown in figure 8, summarize relevant characteristics of the investigated

basin in the partitioning of the hydrological fluxes. The final value of $\Theta$ is achieved after an initial oscillating period, which is a characteristic of the basin too, due to hourly and daily oscillation, especially in evapotranspiration. Without the knowledge of the asymptotic value of $\Theta$, the proper forward probabilities are not known, not even for time past the actual time $t$, because their definitive form depends on future times. This is overlooked in the past literature to which we owe everything, but, in our opinion, is a conceptual passage which is very useful to fully understand the formalism. As we say in the paper, this does not prevent us from knowing the state variables and the backward probability up to time $t$, as it is required by the fact that the past is known. On practical bases, as we shown in Appendix A, $\Theta$ value stabilises after a finite time. So for practical engineering approximations, we just need to wait for a reasonable elapsed time to have the information we require. This time is, in our example, less than the concentration time of the basin of interest.

**Section 6**

**Line 181: as the authors are focusing on a parcel of water that enters at time $t = \tau$ and exits at $t = \iota$ , I think the equations should be rewritten using instead of t.**

We left t but we added further explanations in the text, specifying that t is the exit time.

**Line 187: the symbol S for eq (34) and following is a bit ambiguous as the symbol S(t) already appears at eq. (5) with a different meaning. Moreover, as the integral is defined from 0 to infinity, it appears to me that $S \to \infty$.**

We changed the notation from S to $V_S$ but we think that the integral is correct since it is considering the entire history of the water particles.

**Please give an interpretation of the marginal pdf's.** This will be added in the revised

versionof the paper. An explanation is given in the response to reviewer #3

**I think the notation could be made more "symmetrical". Why is $p(\tau)$ the marginal pdf of the input and pQ(t) is the marginal probability of the output? Shouldn't they be either $p(\tau)$ and $p(\iota)$ or $p_Q(t)$ and $p_J(t)$?**

We changed it in $p_Q(t)$ and $p_J(t)$.

**Line 197: given the notation in eq. (37), it should be $p_Q(t)$ instead of p(t)**

Changed

**Section 7**

**In section 7 and 8 I found several typos and little errors in the formulas. I am not sure I detected them all, so please carefully revise these sections. As in section 5, I think the authors should not assume by default that our knowledge limits to time $t$, and everything between t and $\iota$ is unknown. That is just a very special case, limited to real time forward modeling.**

We tried to correct all the typos we found. However, we insist that, if we derive probabilities "empirically", i.e. after having measured discharge and evapotranspiration, everything after the actual time (clock time) is unknown. Obviously, in practice, in many situations we do not perform real-time forecasting, but use given time series of past events, so the way we operate is going forward in time as it is needed (for instance

for decently estimating $\Theta(\tau)$. But this works because those events are opportunely remote, and er remain well before our clock time.

The limitation to clock time is actually present also in backward probabilities, which are, as well, not known after the clock time. The backward formalism, however, does not require future information to work, and the probability is completely defined up clock time, which is not the case for forward probabilities.

Differently, if we use approaches where probabilities are assigned, we obviously know any future event. But if we assign only the probabilities for discharge (or one of the relevant processes), still the uncertainty of the partition coefficient remains. This is a well known problem in rainfall-runoff model and is related to the determination of the runoff coefficient.

**Line 226: – t instead of – s**

Corrected

**Line 229: isn't the canonic convolution symbol different?**

Changed

**Line 235: typo: time time**

Changed

**Line 237: typo:** $\tau s$

Changed with "all injection times"

**Section 8**

**Please specify that the balance equation is in terms of substance mass.**

Done

**Line 251: please add the dependence on entrance time at left hand side of the equation**

Added.

**Line 261: I would suggest, for consistency with eq. (53) and (54) to name it $C_Q^i(t)$ note in the table of symbols at the end of the manuscript it is listed $C_Q^i(t)$ ).**

Changed.

**Line 263: I don't understand the expression "it is usually assumed the validity of an integral expression like...". Given the definitions of the terms involved in the equation, the integral is just an expression of mass conservation.**

We changed in: "mass conservation implies"

**Line 265: should be a function of $\Theta(\tau)$. Please note that here you use the product $\Theta(\tau)p(t-\tau|t)$ but in the subsequent equations $\Theta(\tau)$ disappears.**

Right, there was a mistake. We correct it in the revised version.

**Line 271: shouldn't it be $p_Q(t - \tau|t)$ instead of $p(t - \tau|t)$ ? In your notation, the latter denotes the age probability of the water storage (eq. 10).**

Corrected accordingly the reviewer suggestions.

**Line 287 and 294-296: why you did not use a Dirac delta function (like in eq. 17) for the age distribution of precipitation?**

Corrected accordingly the reviewer suggestions.

**Line 291-293: this sounds interesting but I am not sure it is actually feasible. I can't say what the issue may be from a mathematical point of view, but I have two examples in mind which are in conflict with the possibility of deriving $p(t - \tau|t)$ and $\omega(t, \tau)$ by coupling different budget equations. The first is that I am not sure the added mass budget equation is actually bringing different information with respect to the water budget equation. If the solute is conservative, it is transported just like water, so its concentration will simply be proportional to that of water, where $C_J^i(t) = 1$. The second is that in case of multi-solute information one could get a system with 3 or more equations and just two unknown functions, so following your reasoning it would be impossible to solve, and this sounds strange to me.**

The main point here is that the SSF cannot be imposed arbitrarily but is computed coupling the the water and solute budgets. What the reviewer asks is if equation (14) and (59), after omitting evapotranspiration in the first, are actually the same equation. It is easy to see that they are not, except for the trivial case in which solute concentration not only is equal in input, output and storage, but also constant.

The second question does not imply problems. In case of multiple solutes, we are measuring the same quantity with two different tools, like we measure a distance of a moving object from a reference point with a laser gun and with a GPS. The two measures have to coincide up to measurements errors. In practice, if the measure is very error prone (as can be the case), Statistics can help to get from the two measures one improved estimate.

**Section 9**

**I think there is a very important assumption here, which is not explicitly stated. The fact that discharge is a linear function of storage does not automatically mean that each parcel of discharge (from an age-rank point of view) is a linear function of the corresponding parcel in storage. This only happen when a well-mixed (o random sampling) scheme is assumed.**

This is certainly true. Linear storage implies just a time invariant and exponential forward distribution. In the paragraph it is actually shown, e.g. equation (69), that backward probabilities are not time invariant.

**I think it is also important that the authors acknowledge here the difference between the storage which can be estimated from a purely hydrologic balance (i.e. the "active" storage which gets displaced during the hydrologic response) and the total storage of a system.**

We are interested here in the formal part of the problem. This is completely, we believed, pretty well explained, especially after the clarification came from reviewers' comments. As we present it (but this is what we understood from the Benettin's himself dissertation) probabilities are derived from the water budget, and the SSF. It can be that
representing the hydrologic response unit (HRU) as a single storage would be insufficient for representing reality with fidelity (e.g. Kirchner, 2003). In this case the single storage equation (9) must be enhanced by adding further storages. This would imply a generalization of our approach that would be straightforward, and, in substance, was already pursued in various papers (e.g., McMillan, 2012; Hrachowitz, 2013). We are pretty aware of it, and we will do some additions in this sense, in the revised version of the paper.

**This is very important considering the recent debate about water displacement and water travel times (e.g. McDonnell and Beven, 2014, Rinaldo et al., 2015, Kirchner, 2016). If the equations derived in this section do not take into account the so called "passive" storage (Kirchner, 2009), they may be of limited practical use**

We already this topic in the answer to first reviewer, and the first author wrote this blog post: http://abouthydrology.blogspot.it/2016/06/celerity-vs-velocity.html to further clarify the issue. The passive storage, in case, has to be present in the water budget equations. If this is accomplished, what it causes is translated in probabilities by the algebra we described.

**Conclusions**

**Line 394: typo "to the obtain understanding"**

Corrected accordingly the reviewer suggestions.

**Appendix A**

**Please define** $\omega(t, \tau)$ **, as only** $\omega(\tau)$ **has been defined so far, and use consistently the upper-case or lower-case symbols. Also, I guess some SSF for both discharge and evapotranspiration was imposed to produce the figures and I think it should mentioned.**

Corrected accordingly the reviewer suggestions.

**References**

Benettin, P., Rinaldo, A., and Botter, G. (2015). Tracking residence times in hydrological systems: forward and backward formulations, Hydrological Processes.

Benettin, P., Rinaldo, A., and Botter, G. (2013). Kinematics of age mixing in advection-dispersion models, Water Resources Research.

McDonnell, J. J., & Beven, K. (2014). Debates - The future of hydrological sciences: A (common) path forward? A call to action aimed at understanding velocities, celerities and residence time distributions of the headwater hydrograph, Water Resources Research.

Kirchner, J. W. (2003). A double paradox in catchment hydrology and geochemistry. Hydrological Processes, 17(4), 871–874. http://doi.org/10.1002/hyp.5108

Kirchner, J. W. (2009). Catchments as simple dynamical systems: Catchment characterization, rainfall-runoff modeling, and doing hydrology backward, Water Resources Research.

Kirchner, J. W. (2016). Aggregation in environmental systems - Part 2: Catchment mean transit times and young water fractions under hydrologic nonstationarity, Hydrology and Earth System Sciences

Dagan, G. (1984). Solute transport in heterogeneous porous formations, Journal of Fluid Mechanics

Ginn, T. (1999). Distribution of multicomponent mixtures over generalized exposure time in subsurface flow and reactive transport: Foundations, and formulations for groundwater age, Water Resources Research

Delhez, E. J. M., Campin, J.-M., Hirst, A. C., & Deleersnijder, E. (1999). Toward a general theory of the age in ocean modelling, Ocean Modelling

Rinaldo, A., Benettin, P., Harman, C. J., Hrachowitz, M., McGuire, K. J., van der Velde, Y., et al. (2015). Storage selection functions: A coherent framework for quantifying how catchments store and release water and solutes. Water Resources Research, 51(6), 4840–4847. http://doi.org/10.1002/2015WR017273

---

## Author Response (AR1)

Trento, September 7, 2016

Dear Editor,

we prepared the revised version of the manuscript "Age-ranked hydrological budget and a travel time description of catchment hydrology" accordingly to our answers to reviewers. Detailed, point to point answer to their observation was already submitted.

The manuscript's Introduction and Conclusions were completely rewritten, a broader context was given to the paper, and its novelties highlighted.

Answering to reviewers implied an enhancement of the paper formalism, by introducing explicitly a the time interval over which precipitations are considered. This made more solid the assertion that Niemi's relation can be interpreted as an expression of the Bayes theorem.

A couple of equations in the section on life expectancy were found incorrect and modified.

To answer to some question raised by the first reviewer, a little historical appendix (B) was added where we discusses ancestors and relatives of our theory. We hope that it could be useful to give the right perspective to our contribution. As requested more or less explicitly by all the reviewers, we added an Appendix (C) where we work out the generalisation of the theory to include many reservoirs.

During the process of revision, we realised that a further aspect could have been emphasised regarding the structure of the backward probabilities, and we added Appendix D, to remark it.

Overall, the main structure of the paper and its other achievements were not modified.

Please find all the modifications from the original submission coloured in red.

We thanks all the reviewers for their constructive and informative comments, and thank you for your work.

For the authors,

Riccardo Rigon

---

## Referee Report (RR1)

**Review of "Age-ranked hydrological budgets and a travel time description of catchment hydrology", by Rigon et al.**

**General Comments**

As pointed out in my previous review, I think this is an interesting article that offers a complementary view on the theoretical foundation of hydrologic transport at catchment-scale. The theory is formulated using probability-theory formalism, which helps making the formulation more general. The paper still includes some little faults which should be polished upon publication. I tried to annotate them in the attached pdf. The English form should also be improved, especially in the new parts introduced after the first review. As one of the authors seems to be English mother tongue, I would suggest a final English proofread. I also write below some minor comments which I hope may help the authors to further improve the paper.

I therefore recommend the paper for publication after technical and language corrections are done.

**Minor Comments for the authors**

- I still believe that some more physical interpretation would help the reader. I report here an example from line 151, where $p_Q(t - t_{in}|t)$ is defined as the "pdf of travel time". For this particular example, the authors could e.g. add that it represents the probability that water in the output Q has entered the system in $t_{in}$.

- Meaning of clock-time $t$: I personally find the separation between the "past" which is completely known and the "future" which is completely unknown a bit too extreme and distant from applications. If one has twenty years of hydrologic data from the year 1990 to 2009, all the forward and the backward distributions that can be estimated over that period will be truncated. However, many of them are likely to be determined up to a satisfying point.

- I think an important clarification should be included in the application to the IUH case (Section 10), as it's been fully clarified in the literature that the IUH describes how a catchment reacts to a precipitation event, but it does not describe the actual time that water takes to travel across a catchment.

- I think it should be made clearer that a linear relationship between the bulk discharge and the bulk storage $Q(t) = \frac{1}{\lambda}S(t)$ does not imply that each single rank storage is proportional to each single rank volume, as it only implies: $\int_0^{min(t,t_p)} q(t,t_{in})dt_{in} = \frac{1}{\lambda}\int_0^{min(t,t_p)} s(t,t_{in})dt_{in}$

- In section 9 the authors stress that the SAS functions "should be derived rather than arbitrarily imposed". This would be true if $C_Q$ and $C_J$ were analytical functions (and the system of equations were easy to solve), but in reality they are data collected at some (often coarse) frequency. It could be at least mentioned that a feasible alternative is that of *testing* possible shapes of the SAS functions against the available tracer data, which in the end is a numerical way to couple the age budget equation to the discharge concentration equation $C_Q(t) = \int C_J(t_i)\,\omega_Q(t - t_i|t)\,p_S(t - t_i|t)\,dt_i$. I think this testing is very different from "arbitrarily assuming" their shape.

[revised manuscript text omitted]

---

## Author Response (AR2)

**Answer to Reviewers regarded the paper: "Age-ranked hydrological budgets and a travel time description of catchment hydrology**

**Answers to Dr. Markus Hrachowitz**

**The authors did a commendable job in addressing my previous comments. The contribution of the manuscript to literature is now much clearer and by putting the analysis into a wider context I am sure it will reach a extended audience. I really enjoyed reading the clarifications and the revised manuscript and recommend publication as is.**

We thank Dr. Hrachowitz for his comment. His review helped us to refocus our manuscript, and we are happy that he enjoyed reading it.

**I have only one small comment concerning the authors' response (11) to my earlier comments: I maintain that in figure 9 of the mentioned article the forward TTDs are shown and, as can be seen, their CDF-proxy does \*NOT\* add up to 1. This highlights, in my understanding, exactly the authors' argument that at a given time t the full forward TTD (i.e. the distribution of times until a given input signal has completely left the system) is not known.**
**best regards, Markus Hrachowitz**

If we properly understand the comment (note that there is no Figure 9), Dr. Hrachowitz is correct. The comment, however, does not imply any change in the manuscript.

**Answers to Dr. Paolo Benettin**

**General Comments**

**As pointed out in my previous review, I think this is an interesting article that offers a complementary view on the theoretical foundation of hydrologic transport at catchment-scale. The theory is formulated using probability-theory formalism, which helps making the formulation more general. The paper still includes some little faults which should be polished upon publication. I tried to annotate them in the attached pdf. The English form should also be improved, especially in the new parts introduced after the first review. As one of the authors seems to be English mother tongue, I would suggest a final English proofread. I also write below some minor comments which I hope may help the authors to further improve the paper. I therefore recommend the paper for publication after technical and language corrections are done.**

We thank Dr. Benettin for his review. His observations were very constructive and detailed, and we used all of them. We have edited the writing style and English grammar in the new manuscript, and provided answers to each of his new comments.

**Minor Comments for the authors**

**I still believe that some more physical interpretation would help the reader. I report here an example from line 151, where $p_Q(t - t_{in}|t))$ is defined as the "pdf of travel time". For this particular example, the authors could e.g. add that it represents the probability that water in the output $Q$ has entered the system in $t_in$**

Added

**Meaning of clock-time $t$: I personally find the separation between the "past" which is completely known and the "future" which is completely unknown a bit too extreme and distant from applications. If one has twenty years of hydrologic data from the year 1990 to 2009, all the forward and the backward distributions that can be estimated over that period will be truncated. However, many of them are likely to be determined up to a satisfying point.**

True. However, this has to do with the need to use historical (past) data in papers. For real-time applications, the perspective changes, and our view is a necessity.

**I think an important clarification should be included in the application to the IUH case (Section 10), as it's been fully clarified in the**

**literature that the IUH describes how a catchment reacts to a precipitation event, but it does not describe the actual time that water takes to travel across a catchment.**

Good observation. The clarification is added in the new text. However, it occurs in section 5, not in section 10.

**I think it should be made clearer that a linear relationship between the bulk discharge and the bulk storage $Q(t) = \frac{1}{\lambda} S(t)$ does not imply that each single rank storage is proportional to each single rank volume, as it only implies: $\int_0^{\min t, t_p} q(t, t_{in}) dt_{in} = \frac{1}{\lambda} \int_0^{\min t, t_p} s(t, t_{in}) dt_{in}$**

Dr. Benettin is right that mathematically the linear relation of the whole system does not imply an equivalent linearity for the subsystems. In fact,

$$\int_0^{\min (t, t_p)} \left( q(t, t_{in}) - \frac{1}{\lambda} s(t, t_{in}) \right) dt_{in} = 0 \tag{1}$$

just implies that the integrand would be either null, or an oscillating function (in $t_{in}$) which integrates to zero in $[0, \min(t, t_p)]$. To obtain this, however, it is easy to show that the subsystems' lambdas should become time variant, and must be chosen accurately "ad hoc" to nullify the integrand for any clock time $t$. Therefore the linear bulk system as obtained by linear subsystems remains the simpler option. We added in the text: "assuming that the age-ranked storages behave linearly ...".

**In section 9 the authors stress that the SAS functions "should be derived rather than arbitrarily imposed". This would be true if $C_Q$ and $C_J$ were analytical functions (and the system of equations were easy to solve), but in reality they are data collected at some (often coarse) frequency. It could be at least mentioned that a feasible alternative is that of testing possible shapes of the SAS functions against the available tracer data, which in the end is a $C_Q(t) = \int C_J(t_{in}) \omega_Q(t - t_i) p_S(t - t_i | t) dt_{in}$ I think this testing is very different from "arbitrarily assuming" their shape.**

We dropped the word "arbitrarily". However, equations involving concentrations, do not require knowledge of $C_Q()$, but of the injection and storage concentrations, which are assumed to be known in all the approaches.

**Notes in the paper**

**Page 2 - line 46 - There is something wrong in this phrase: "A more detailed history of these concepts can be found in Benettin et al. (2013) and Hrachowitz et al. (2016) and Appendix B, is more specifically related to this paper"**

The phrase was modified: "A more detailed history of these concepts can be found in Benettin et al. (2013) and Hrachowitz et al. (2016). Appendix B includes a brief review more specifically related to the topic of this paper.

**Page 5 - Equation 9. I think it would be useful to explain in words what the interpretation of eq (9) is (especially because the derivative d/dt is not developed into partial derivatives, so the interpretation is slightly different from van der Velde's and Harman's equations).**

To the phrase: "These equations were introduced first by van der Velde et al. (2012) and named by Harman (2015a), even if similar ones were present in previous literature, as discussed in Appendix B" we added: "In our formulation, however, by using $t$ and $t_{in}$, instead of $t$ and $T_r$, as independent variables we do not need to transform the original ordinary differential equations into partial differential equations."

**Page 6 -line 143 - eq. 10 - Yes, but what is this $p$ ?**

We changed it to $p_S$, as it should be.

**Page 6 -line 187 - Yes, but what does this physically mean? I am afraid the reader would have a hard time understanding this.**

What is to be understood is that this is not a pdf in t, but just in $t_{in}$. This fact remained kind of unclear in previous treatments of the matter, because they did not use conditional probabilities. This is actually explained in the lines below 187 (first revised manuscript).

**Page 8 - Figure 2 - Why does the x-axis is "injection time"? Indeed, the y-axis label says that ps is a function of t-tin, i.e. the residence time.**

The the domain of the pdfs (in this case backward residence time pdfs) is bidimensional: one dimension is the injection time (given by the time series of the inputs), the other one is the actual time (how each input evolves with the actual time). If we consider to have the injection times in rows and the actual times in columns, Figure 2 is obtained by plotting all the results in the same column ($t_{in}$ varies and $t$ is fixed) while Figure 4 is obtained by plotting all the results for the single row ($t_{in}$ is fixed and $t$ varies)

**Page 8 - line 196 - I am not sure this new notation with the subscript S is appropriate here and in the following paragraphs. This variable does not describe a property of the storage. It is just the (cumulative) probability that a precipitation input entered at a past**

**time ti will exit the system at time t (either through Q or ET)**

It is a property of the storage, since it is a function of the storage age-ranked functions. This applies in the whole paragraph.

**Page 11- Caption of Figure 5 - Forward distribution of what?**

Residence time forward distribution.

**Page 12 - please mention which SAS is used for the simulation**

We specified in the text that we were considering the complete mixing case $(\omega_Q(t, t_{in}) = \omega_{E_T}(t, t_{in}) = 1)$.

**Page 16 - I think this is the variable that should be defined as pS, for consistency with the backward framework (eq 10)**

We changed with $p_S$ everywhere.

**Page 16 line 299 - this variable is not defined and does not appear in the notation appendix**

It should be: $p_Q = p_{t_{ex}} * p_S$. However, because this passage is not very informative we deleted the whole sentence from line 297 to line 300.

**Page 16 - line 308. Unclear**

We changed the phrase as follows: The variable $t_k$, used for making compact equations above and below, is such that $t_0 = t_{ex}$ ($k = 0$) and $t_1 = t$ ($k = 1$).

**Page 18 - please make it clearer that the perfect mixing just refers to the water sample (to make sure the reader cannot think the entire catchment is well mixed)**

We changed with: "When a sample is taken, the action implies perfect mixing of all the age-ranked waters in the container where measurements are made."

**Page 19 - this title is a bit unclear**

We changed the title to: "A simple example where probabilities are assigned instead of derived."

**Page 19 - this is prone to misunderstandings, as the "mean travel time" in the IUH framework actually refers to the response time of a catchment (which involves the displacement of old water) and not to the actual water travel times. If one estimates $\lambda$ from hydrological measurements, the actual water travel time would be underestimated by orders of magnitude.**

We changed "mean travel time" to "mean response time" and specified that it refers to forward distributions, not to backward ones.

**Page 20 - this is not a results. You have assumed a uniform SAS function from the beginning!**

Not true. We are not assuming SAS here. Instead, we are assigning the forward storage probabilities. Thus SAS values result from this assignment.

**Page 21 - but can't $ae(t, t_{in})$, in general, depend on $s(t, t_{in})$ ? Then eq. (77) would not be linear anymore.**

This discussion can be applied to eq. (76). Obviously the dependence on $ET$ could be made non-linear, and it probably is at least slightly non-linear. In that case, solutions have to be found numerically. However, all the cases made by Dr. Benettin in his papers, to our knowledge, are linear.

**Page 27 - I think your approach is "spatially integrated" rather than "coarse grained". You can actually derive the budget equation by integrating the equations of the local approach.**

In the revised manuscript, we changed the text to "spatially integrated".

**Page 29 - too vague as a sentence. Which "experimental results" do the authors talk about? I think the cited paper does the opposite: it shows how one single reservoir can be used to properly model discharge.**

Dr. Benettin is correct about the citation to this specific Kirchner paper, but, since it is unessential to our arguments, the citation have been dropped. We rephrased it as follows: "Looking at the literature we cited in the main text, it is usually recognized that a single reservoir is not able to reproduce proper discharge and tracer behavior. Usually a few "embedded" reservoirs are deployed in models."